# Twistedly hydrophobic basis with suitable aromatic metrics in covalent organic networks govern micropollutant decontamination

Chencheng Qin[1,5], Yi Yang[1,5], Xiaodong Wu[2,5], Long Chen[3], Zhaoli Liu[3], Lin Tang [1], Lai Lyu[4], Danlian Huang[1], Dongbo Wang[1], Chang Zhang[1], Xingzhong Yuan[1], Wen Liu [3] ✉ & Hou Wang [1] ✉

The pre-designable structure and unique architectures of covalent organic frameworks (COFs) render them attractive as active and porous medium for water crisis. However, the effect of functional basis with different metrics on the regulation of interfacial behavior in advanced oxidation decontamination remains a significant challenge. In this study, we pre-design and fabricate different molecular interfaces by creating ordered π skeletons, incorporating different pore sizes, and engineering hydrophilic or hydrophobic channels. These synergically break through the adsorption energy barrier and promote inner-surface renewal, achieving a high removal rate for typical antibiotic contaminants (like levofloxacin) by BTT-DATP-COF, compared with BTT-DADP-COF and BTT-DAB-COF. The experimental and theoretical calculations reveal that such functional basis engineering enable the hole-driven levofloxacin oxidation at the interface of BTT fragments to occur, accompanying with electron-mediated oxygen reduction on terphenyl motif to active radicals, endowing it facilitate the balanced extraction of holes and electrons.

Water crisis caused by emerging microcontaminants including pharmaceuticals and endocrine-disrupting chemicals poses great threats to human health, due to their potential toxicity, persistence, and low biodegradability[1–5]. To enable a carbon-neutral industrial society, utilizing sustainable and clean solar-energy photocatalysis to meet the increasing energy demand has triggered broadened interest for environmental remediation, but it still exists challenges[6,7]. Considering that the potential release of metal ions from metal-based photocatalyst into the water could produce secondary contamination, nonmetallic reticular materials-based catalysis is regarded as alternative method to accelerate the wastewater purification[8–10]. However, the fact that how to achieve efficient photochemical decomposition of pollutants while reducing energy consumption in water treatment has been long ignored. Up to now, quite a limited strategy has been proposed for simultaneous regulation of electronic structure and interfacial reaction from the views of functional basis in the solid-liquid interface.

Heterogeneous photochemical decomposition of organic pollutants includes surface adsorption, electron-hole separation

[1]College of Environmental Science and Engineering and Key Laboratory of Environmental Biology and Pollution Control (Ministry of Education), Hunan University, 410082 Changsha, China. [2]College of Materials Science and Engineering, Nanjing Tech University, 210009 Nanjing, China. [3]The Key Laboratory of Water and Sediment Sciences, Ministry of Education, College of Environmental Sciences and Engineering, Peking University, 100871 Beijing, China. [4]Institute of Environmental Research at Greater Bay Area; Key Laboratory for Water Quality and Conservation of the Pearl River Delta, Ministry of Education, Guangzhou University, 510006 Guangzhou, China. [5]These authors contributed equally: Chencheng Qin, Yi Yang, Xiaodong Wu. ✉e-mail: wen.liu@pku.edu.cn; wangh@hnu.edu.cn

(generation of active species), and product detachment and transport (regeneration of active sites). The second step is intimately tied to the electronic structure, while the first and third steps are respectively governed by surface characteristic[11]. Firstly, adsorption is the vital procedure during the surface/interface-mediated photochemical reaction. The key to overcoming the adsorption barrier between the contaminants and the surface energy of catalyst is to create an appropriate surface microstructure, which may realize energy savings and consumption reduction at the macro-level of water treatment. Secondly, the close proximity between hole-driven reaction sites and electron-driven reaction sites could enable the oxidation and the reduction to occur in series[12]. Nevertheless, achieving local separation and inhibiting recombination at only nanoscale intervals is particularly challenging. Finally, the separation and conveyance of the degraded products is the indicator of the catalyst's capacity to self-purify. In turn, renewing the number of active sites on the catalyst surface accelerates the pollutant removal. However, regulating the overall factors of photochemically removing pollutants in combination with the above three perspectives is a big challenge.

As a type of porous crystalline polymers, covalent organic frameworks (COFs) with a hierarchical lamellar structure and one-dimensional open channel could shorten the transfer distance of electrons to surface active sites in the periodic framework, enabling the rapid charge diffusion to trigger the heterogeneously photochemical reaction[13–17]. Most crucially, the structural tunability of COFs easily realized by practical building blocks with abundant topologies and dimensionalities. In order to increase the electron transport and the electrophilic attack of free radicals in organic pollutant mineralization, it is necessary to introduce electron-rich building blocks into COFs. Thiophene is a stable π-aromatic five-membered ring compound, and the n−π* transition of the long pairs on sulfur and suitable molecular orbital occupancy gives it excellent conductivity and adjustable electron density[18,19]. The sulfur-rich benzotrithiophene (BTT) has a planar conjugated system of $C_{3h}$ symmetry, where three thiophene rings are blend on the benzene central ring, tending to achieve π-electron delocalization and high conjugation effect[20]. It is manifested in two aspects. On the one hand, the extended π-conjugated framework yields a narrow band gap that broadens the energy-harvesting capability to a wide visible-light absorption spectrum. On the other hand, the π networks enable exciton migration and facilitate forward electron transfer and charge-carrier transport to prevent backward charge recombination[21,22]. For the exploration of porous COF platforms to challenge water issues, the limitation of mass transfer and reactive sites exposure must be overcome based on percolation theory and inner-surface renewal theory. The answers to the issues are without a doubt found in intrinsic channel and surface microstructure. In terms of microporous COFs, mismatch between molecules and pore sizes make it difficult for substrates to enter the channel and for transformed products to leave the channel[23]. By altering the function motifs, the hydrophobicity can be adjusted to enhance the selectivity of adsorbing pollutants and optimize the catalytic performance in the surface/interface of the water catalyst. The aromatic with a twist in shape still transforms the electronic properties of the molecule, including the overall electronic change of the organic skeleton. Therefore, a subtle structural variation for COFs–especially in the interface, molecular ordering and organic building blocks–can completely change the overall performance toward an energy-saving and cost-reducing way.

Hence, a systematic interfacial design of photocatalysts was successfully explored. We used BTT as the electron-donated building block and introduced aromatic aniline with different distortions, including 1,4-diaminobenzene (DAB), 4,4-diaminodiphenyl (DADP), and 4,4-diaminoterphenyl (DATP) to modulate the electronic structure, pore size and surface feature of the framework, which

synthesized three kinds of imine-linked COFs and termed as BTT-DAB-COF, BTT-DADP-COF, and BTT-DATP-COF (Fig. 1a). Through the thoughtful optimization of building pieces, the surface/interface performance of pollutant removal by COF was established. Specifically, the surface adsorption energy of the novel BTT-DATP-COF for pollutants is greater ~8 times (−3.02 eV) than other COFs, and the removal capacity of levofloxacin hydrochloride (LEV) under visible light irradiation is 6 and 8 times that of BTT-DADP-COF and BTT-DAB-COF. The photochemical decomposition capacity of COFs toward typical antibiotics in wastewater is universal. The difference in the three COFs on the intrinsic electron structure and photo-excited charge transfer is confirmed by the transient absorption spectroscopy (TA) and density function theory (DFT). Additionally, functional basis engineering with various metrics allowed the holes to drive LEV oxidation at the interface of the BTT fragment, along with electron-mediated reduction of triphenyl-sequence oxygen to active free radicals (i.e., dissolved oxygen in water as an electron and energy acceptor), which was helpful in promoting balanced extraction of holes and electrons and reducing external energy consumption.

## Results
### Configuration and properties of the COFs
The unit cell paraments and crystallinity of prepared COFs were confirmed by power X-ray diffraction (PXRD) measurements, and the Pawley refinement demonstrated the good fit of the eclipse stacking model (AA stacking) for three COFs (Fig. 1b, c). The unit cell paraments of new BTT-DATP-COF ($a = b = 44.49 Å$, $c = 3.51 Å$, $α = β = 90°$, $γ = 120°$) were obtained in Supplementary Table 2. All of BTT-DAB-COF, BTT-DADP-COF, and BTT-DATP-COF have high crystallinity with the correspondingly respective (100) plane of 3.50°, 2.78°, and 2.26°, respectively (Supplementary Fig. 2). The peaks of BTT-DAB-COF and BTT-DADP-COF are consistent with previous work[24]. After normalizing the PXRD intensity (Supplementary Fig. 1), the (100) plane peak of BTT-DAB-COF, BTT-DADP-COF, and BTT-DATP-COF exhibit full width at half maximum (FWHM) values of 0.47°, 0.31°, and 0.25°, respectively. Additionally, the corresponding microcrystal size is calculated as 16.80 nm, 25.88 nm, and 32.30 nm, respectively. According to the *Scherrer* equation, the sharp reflection, narrow half-peak width (FWHM), and large crystal size indicate a large ordered domain and very low concentration of defects, meaning that the BTT-DATP-COF has the maximum π-conjugated and ordered degree[25,26]. The wide peak at 25.21°, 25.96 °, and 26.44° can be assigned to the (001) plane generated by π stacking of 2D layers for the three COFs (Supplementary Fig. 2). The shifted toward higher degree for BTT-DATP-COF indicates a decreased distance of interlayer stacking and better degree of longitudinal conjugation. Furthermore, DAB, DADP, and DATP building blocks have torsion angles via measuring the dihedral angles using adjacent four atoms, corresponding to 28.5°, 145.8° and 146.9° respectively (Fig. 1a and Supplementary Movie 1, 2, and 3). The trend in torsion angles as derived from single crystal data concurs with the trend discerned from computational calculations. The corresponding single crystal data are 22.740°, 26.085° and 27° for DAB (CSD number: 1828333) DADP (and CSD number: 625724) DATP (CSD number: 1269383), respectively. Although triphenyl has the greatest torsion angle, it is a flexible monomer that has been shown to be more efficient at stacking during polymer reactions[27]. Reversible bond formation and structural self-healing have a central role in achieving long-range crystalline order for COFs[28]. In the presence of reversible imine linkage, the more torsion of the DATP building block than that of DAB and DADP probably strengthens the structural self-healing of COFs during the dynamic polymerization between BTT and aromatic aniline, thus achieving more long-ranged order COFs. This is consistent with the result that the order of crystallinity for BTT-DAB-COF < BTT-DADP-COF < BTT-DATP-COF. It verifies that the difference in torsion angle in

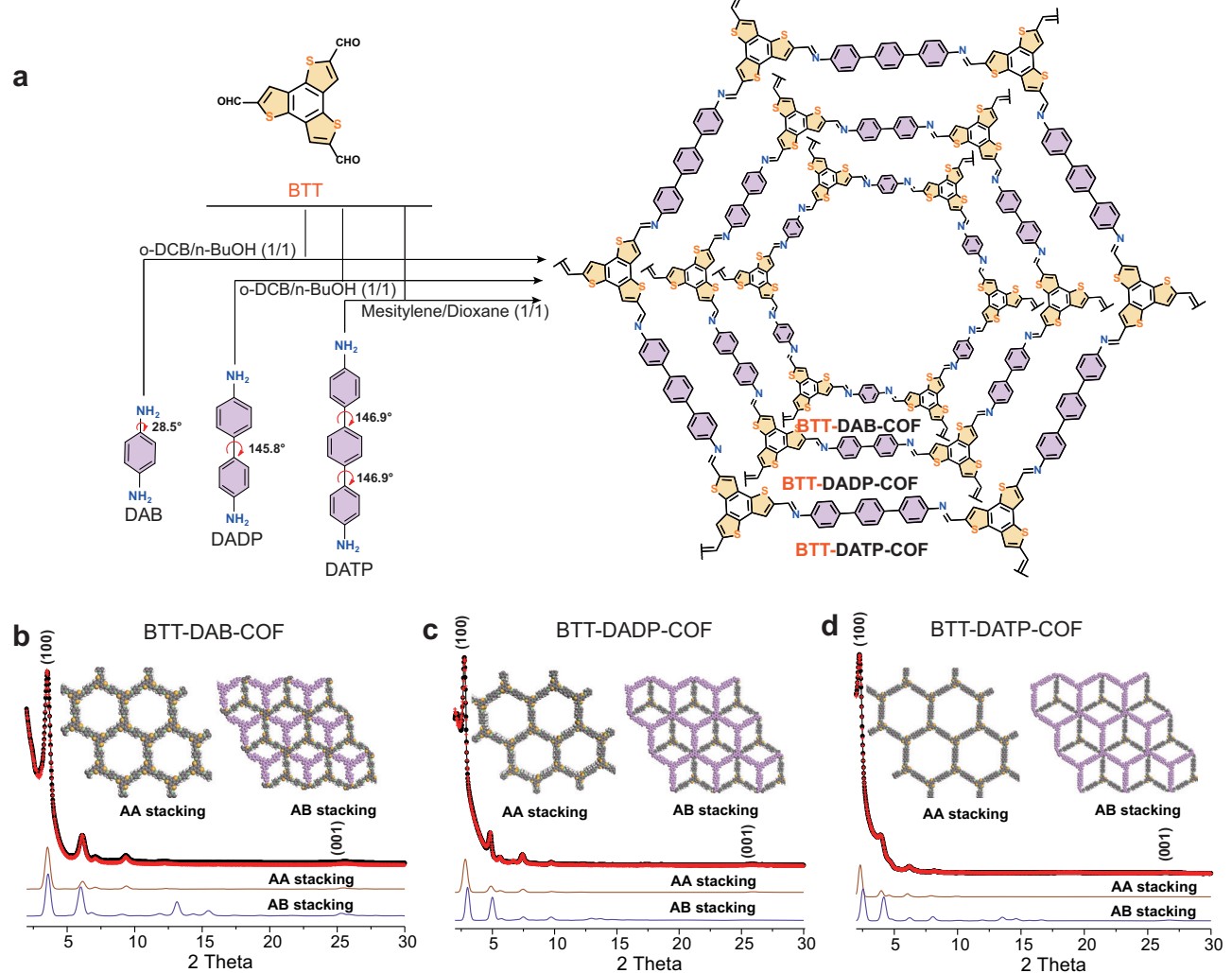

**Fig. 1 | Synthetic method of BTT-based COFs with torsion angles and stacking.**
**a** Schematic represented of the syntheses of BTT-based COFs with different
pore size, termed as BTT-DAB-COF, BTT-DADP-COF, and BTT-DATP-COF.
**b–d** Experimental, refined and simulated PXRD patterns of BTT-DAB-COF,
BTT-DADP-COF, and BTT-DATP-COF.

linkers varies the crystallinity, conjugation, and long-range order of
COFs. Besides, Fourier transform-infrared spectroscopy (FT-IR) spec-
tra and X-ray photoelectron spectroscopy (XPS) confirmed the suc-
cessful synthesis of all three types of COFs via Schiff-base
condensation polymerization (Supplementary Figs. 3, 4).

The porous structure of COFs was verified by the nitrogen
adsorption-desorption measurements at 77 K (Supplementary Fig. 5).
Three COFs exhibited the typical IV isotherm with an H3 hysteresis
loop, which confirmed the presence of mesoporous structure. The
specific surface area, pore volume, and average pore size are shown in
Supplementary Table 3. The pore size of BTT-DAB-COF, BTT-DADP-
COF, and BTT-DATP-COF corresponded to 2.2, 2.7, and 3.2 nm in
Fig. 2a, respectively, which aligns with the decreasing trend observed
for the diffraction angles at the (100) plane. This further proves that
the pore size of BTT-based COFs rises with the increase of the number
of benzene rings in the phenyl, biphenyl, and terphenyl linker. The
specific surface area (SSA) of BTT-DAB-COF, BTT-DADP-COF, and BTT-
DATP-COF, calculated by the Brunauer–Emmett–Teller method, cor-
responded to 2372, 1676, and 1340 m²/g, respectively (Supplementary
Table 3). Relatively, BTT-DATP-COF with maximum pore size and pore
volume may promote more exposure to active sites and the mass
diffusion, involving organic molecule and active radical species.
Thermogravimetric analysis (TGA) results demonstrated that BTT-

DAB-COF materials are thermally stable up to 400–500 °C under the
protection of N₂, while BTT-DADP-COF and BTT-DATP-COF begins to
decompose at ~300 °C (Supplementary Fig. 6). The decomposition
rate follows the order of BTT-DATP-COF > BTT-DADP-COF > BTT-
DAB-COF.

The morphologies and microstructure of the three COFs were
characterized by Scanning/transmission electron microscopy (SEM/
TEM). For BTT-DAB-COF, BTT-DADP-COF, and BTT-DATP-COF pos-
sessing the phenyl, biphenyl, and terphenyl units, the morphology
changed from irregular short dendritic crystal materials (Supplemen-
tary Fig. 7a) to sea urchin-like material covered by petal-like nanorods
(Supplementary Fig. 7b), and finally to elongated smooth nanowire
structures with a diameter of around 100 nm (Supplementary Fig. 7c).
By introducing malleable long-chain aromatic benzene ring, the soft
fiber structure of BTT-DATP-COF can intertwine with each other,
forming large reticular channels which are conducive to the full con-
tact of the reactant to COFs[29,30]. Besides, the high-resolution TEM
(HRTEM) images exhibited the interlayer stacking of 3.9, 3.6, and 3.3 Å
along the *c* direction for respective BTT-DAB-COF (Supplementary
Fig. 8a), BTT-DADP-COF (Supplementary Fig. 8b) and BTT-DATP-COF
(Supplementary Fig. 8c), corresponding to the (001) plane. The trend
of wide peak value from the PXRD pattern is negatively correlated to
the interlayer stacking exhibited by the high-resolution TEM (HRTEM)

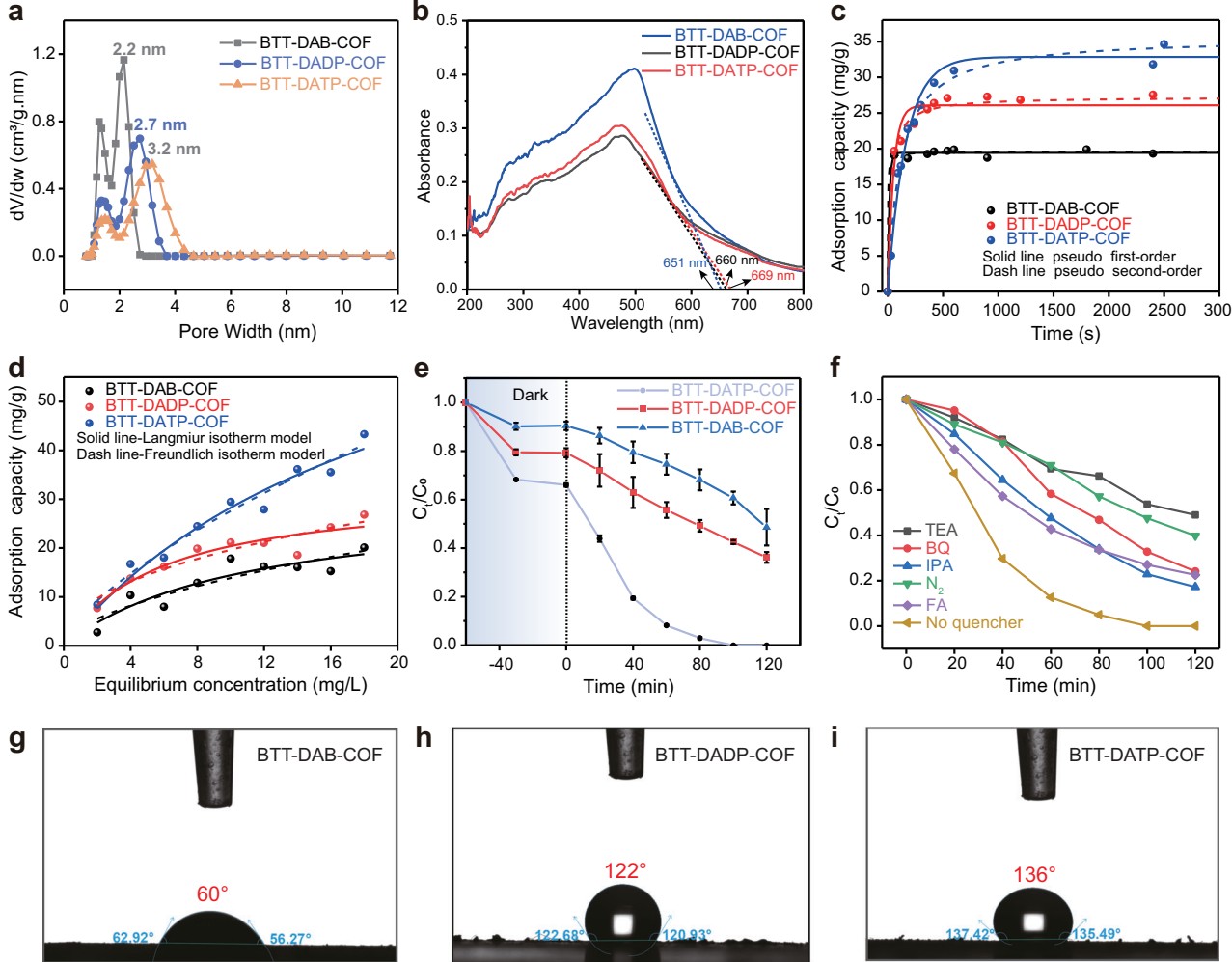

**Fig. 2 | Comparison of properties and pollutant degradation of COFs. a** Pore distribution of three COFs. **b** UV-vis DRS pattern of three COFs. **c, d** Adsorption kinetics and isotherms recorded for LEV onto three COFs. **e** Photocatalytic activity for LEV degradation by different COF under visible-light illumination. **f** Photocatalytic performance of COFs with different scavengers under equivalent reaction conditions. **g–i** Contact angles of BTT-DAB-COF, BTT-DADP-COF, and BTT-DATP-COF.

images, indicating a better degree of longitudinal conjugation for BTT-DATP-COF[31].

By the UV–visible diffuse reflectance spectrum (UV–Vis DRS) in Fig. 2b, three COFs have an absorption onset at 651, 660, and 669 nm, respectively. The corresponding energy band gaps (Eg) decreased from 2.21 to 2.18, and 2.15 eV (Supplementary Fig. 9). This suggests that terphenyl units enhance the conjugation effect across the plane (along the c plane of the z-axis) of the imine-linked BTT-based COFs, resulting in a narrower band gap, which is favorable for electronic transitions[32]. A donor-acceptor (D-A) torsional spring between BTT and aromatic aniline (DAB, DADP, or DATP) undergoes force-induced planarization during uniaxial elongation leading to red-shifted absorption[33]. This is consistent with the result that BTT-DATP-COF has the highest crystallinity, as well as the result of (001) crystal plane shift. On the basis of ultraviolet photoelectron spectroscopy (UPS) (Supplementary Fig. 10), the relative valence band maximum (VBM) of BTT-DAB-COF, BTT-DADP-COF and BTT-DATP-COF is calculated to be 2.19, 2.33, and 2.37 eV (vs. Relative Hydrogen Electrode-RHE)[34]. We further carried out the Mott Schottky electrochemical measurements to determine the energy band positions (Supplementary Fig. 11), the conduction band (CB) of BTT-DAB-COF, BTT-DADP-COF, and BTT-DATP-COF are calculated to be −0.10 eV, −0.03 eV, and 0.02 eV (vs. RHE) and −0.45, 0.4 and −0.35 eV (vs. Normal Hydrogen Electrode-NHE), respectively. Meanwhile, the VB positions of BTT-DAB-COF, BTT-DADP-COF, and

BTT-DATP-COF are 2.11 eV, 2.15 eV, and 2.17 eV (vs. RHE), respectively, which is not much different from the values obtained by UPS. The specific band positions are displayed in Supplementary Fig. 12.

## Performance evaluation toward removal of LEV
Adsorption is the transfer process of pollutants at solid–liquid interface, which is the key procedure during the surface/interface-mediated catalytic reaction. A series of batch adsorption experiments were adapted to investigate the dynamic adsorption process of LEV on COFs. The adsorption capacity of adsorption kinetics and adsorption isotherms by COFs (Fig. 2c, d) followed this order: BTT-DAB-COF < BTT-DADP-COF < BTT-DATP-COF. Specifically, the saturated adsorption capacity ($q_m$) calculated according to Langmuir isotherm model of BTT-DATP COF was 86.65 mg/g, which was much higher than that of BTT-DAB-COF (29.99 mg/g) and BTT-DADP-COF (32.30 mg/g) in Supplementary Table 5. However, the rate constant for second-order adsorption kinetic ($k_2$) of BTT-DAB-COF (0.02551 g·mg/s) was greater than the other two COFs in Supplementary Table 4. All the adsorption kinetic data, analyzed using mass-transfer model and relevant parameters were summarized in Supplementary Table 6 and Supplementary Fig. 13. The calculated mass-transfer coefficient ($k_f$) value for the LEV molecules was $4.2 \times 10^{-4}$, $1.01 \times 10^{-4}$ and $3.88 \times 10^{-5}$ cm/s for BTT-DAB-COF, BTT-DADP-COF, and BTT-DATP-COF, respectively. Obviously, all the results show that the adsorption capacity is

positively correlated with the pore size, independent of the BET-surface area values of COFs. However, the adsorption kinetics and transfer coefficient are inversely proportional to the pore size. Based on this, the wetting property of three COFs was further conducted in Fig. 2g–i. The water contact angle of BTT-DATP-COF (136°) was much larger than that of BTT-DADP-COF (122°) and BTT-DAB-COF (60°), indicating the BTT-DATP-COF has difficult dispersion into water in a short time due to the hydrophobic effect. Due to the steric effect, the small pore size of BTT-DAB-COF would prevent LEV molecules from entering the channels to obtain low accumulated capacity, and quickly reaching the adsorption equilibrium onto the external surface. Since all coefficients are related ($k_2$ and $k_f$) to the time of adsorption equilibrium, BBT-DAB COF has the highest coefficient. The large aperture of BTT-DATP COF allows LEV molecules to be easily adsorbed and transported into the channels of the framework, thus promoting the subsequent cascaded reactions upon the visible light irradiation. Furthermore, the π-π stacking interaction between COFs and LEV, and the physical/morphological properties of COFs after adsorption LEV were investigated in Supplementary Figs. 14–17.

The LEV degradation after adsorption equilibrium by COFs was further evaluated, as displayed in Fig. 2e. The BTT-DAB-COF and BTT-DPDA-COF could remove 51.4% and 73.8% of LEV after 120 min upon visible-light irradiation, while a LEV removal efficiency of 100% (below the detected limitation) could be achieved by BTT-DATP COF within 100 min. Furthermore, almost no photolysis of LEV occurred under visible-light irradiation in the absence of COF materials (Supplementary Fig. 18). To analyze the degradation kinetics of COFs, all of the degradation processes were normalized and fitted by first-order kinetics model, and the results are provided in Supplementary Fig. 19. BTT-DATP-COF exhibited the highest apparent rate constant (0.0394 min$^{-1}$), a 6- and 8-fold increase in the degradation rate compared with BTT-DAB-COF and BTT-DADP-COF, respectively. Besides, the apparent quantum efficiency (AQE) provided by BTT-DATP-COF were evaluated at a different single light wavelength of 450–650 nm, and the material displayed a maximum AQE value of 34% at 450 nm (Supplementary Fig. 20), which is well associated with UV−vis DRS spectra. After enduring four cycles, the removal efficiency of LEV by BTT-DATP-COF remained above 85% (Supplementary Fig. 21), and contained a similar XRD pattern (Supplementary Fig. 22). The developed three COFs showed high chemical stability and photostability even in harsh conditions (strong acid, strong alkaline and water with high temperature), as no organic ligands would dissolve into solution during a long-term test based on LC-MS/MS analysis and total organic carbon (TOC) change (Supplementary Fig. 23). Supplementary Table 7 displays that the performance of BTT-DATP-COF is significantly better than that of previously reported materials on LEV degradation under identical operating conditions. Therefore, the structural superiority of BTT-DATP-COF with excellent properties not only improves the adsorption process of LEV, but also greatly governs its degradation ability.

Specific reactive species with various contribution efficiencies during the degradation process were subsequently identified according to the electron-spin resonance (ESR) spectra and the scavenger quenching experiments. As shown in Supplementary Fig. 24 upon exposure of visible light, all of three COFs revealed detectable DMPO-·O$_2^-$, TEMP-$^1$O$_2$ and DMPO·OH signals under experimental conditions. The longer the extension of light time is, the stronger the signal is. Interestingly, the signal intensity for TEMPO·h$^+$ decreased with increasing irradiation duration, indicating the reaction between TEMPO and photogenerated h$^+$ by COFs. The TEMPO was further oxidized to the TEMPO$^+$ by the photogenerated h$^+$ under visible-light irradiation. The observation confirmed that the disappearance of the TMMPO signal in the ESR spectrum was attributed to the increase of photogenerated h$^+$. Take the BTT-DATP-COF as a typical example in Fig. 2f, when h$^+$, ·O$_2^-$, ·OH and $^1$O$_2$ were eliminated by quench agents,

corresponding to triethanolamine (TEA), p-benzoquinone (BQ), iso-propyl alcohol (IPA) and furfuryl alcohol (FA), the removal efficiency was decreased to 51.1%, 75.9%, 82.8%, and 77.4%, respectively. The corresponding rate constants were displayed in Supplementary Fig. 25. According to Supplementary Method[35,36], the relative contribution of h$^+$, ·O$_2^-$, ·OH, and $^1$O$_2$ to the overall LEV degradation were calculated to be 29.9%, 24.3%, 21.7% and 24.1%. Clearly, the h$^+$ had a relatively significant contribution to the degradation of LEV than that of other reactive species. This phenomenon demonstrated that the highly positive HOMO position of BTT-DATP-COF makes its photogenerated holes possess stronger oxidation abilities. Moreover, the system was filled with N$_2$ to eliminate the influence of O$_2$, and the degradation rate of LEV decreased from 100% to 60.1%. The utilization of both dissolving O$_2$ and solar energy-induced oxidized holes provides an energy-saving and cost-reducing way for water purification.

## Degradation pathway and toxicity assessment of LEV

Figure 3 displays the Fukui index of LEV calculated at the level of B3LYP/6-31 G*[37], which is applied to investigate the reactive sites of LEV during degradation. The electrostatic potential (ESP) distribution shows that the electron-rich region of LEV is concentrated in the carboxylic acid group, while the electron-deficient region is located in the alkyl group (Fig. 3a). The highest occupied molecular orbitals (HOMO) of LEV molecule represent that the regions of C17-N18 in the piperazine ring of LEV has high active ability to lose electrons (Fig. 3b), which prefer to be attacked by electrophilic species such as h$^+$, $^1$O$_2$ and ·OH. In comparison, the lowest unoccupied molecular orbitals (LUMO) show that the regions of C6-C7 have higher active ability to gain electrons (Fig. 3c), which tend to be attacked by nucleophilic species such as ·O$_2^-$. The detailed nature population analysis (NPA) charge distribution values and Fukui index for electrophilic attack ($f^-$), nucleophilic attack ($f^+$), and radical attack ($f^0$) are further calculated and displayed in Supplementary Table 8. Therefore, based on transformation products (TPs) detected by liquid chromatograph-mass spectrometry (LC-MS), four degradation pathways of LEV are proposed in Supplementary Fig. 26 and Supplementary Table 9. In this photocatalysis system, h$^+$, ·OH, ·O$_2^-$, and $^1$O$_2$ are confirmed as the primary reactive species, so the Fukui index of electrophilic attack ($f^-$) and radical attack ($f^0$) are mainly considered. Specifically, N18 ($f^- = 0.2136$, $f^0 = 0.1129$) and N21 ($f^- = 0.0638$, $f^0 = 0.0318$) with high Fukui index are the most active sites for electrophilic and radical attack, indicating that the pathway II (demethylation, product $m/z = 347.1389$) and pathway III (hydroxylation, product $m/z = 363.1566$) should be the primary pathways for LEV degradation. In addition, the C6 ($f^+ = 0.1866$) exhibits the highest reactivity for nucleophilic attack, leading to the hydrolysis of adjacent carboxylic acid groups in pathway IV (product $m/z = 333.1550$) after ·O$_2^-$ attack. The total organic carbon (TOC) removal experiment manifested that BTT-DATP-COF can efficiently mineralize 54% of LEV into inorganic products after 2 h illumination. Furthermore, it was found that the concentrations of fluoride ion (F$^-$) and nitrate nitrogen (NO$_3^-$-N) after the reaction increased by 4.5 and 1.4 mg/L, respectively, while the organic nitrogen (DON) decreased by 0.5 mg/L. Degradation pathway results showed that the intermediate and final products were mostly hydrophilic substances (TP186, TP94, etc.), compared with the parent LEV molecule. The water vapor adsorption isotherms confirmed that BBT-DATP-COF has a hydrophobic pore structure (Supplementary Fig. 27), that is, with a hydrophobic inner surface. Therefore, the hydrophilic by-products are more likely to fall off the hydrophobic surface of BTT-DATP-COF and return to the aqueous phase, releasing the active sites on the surface of the material. With the help of favorable transport channel of BTT-DATP-COF, by-products could be transferred out in time to further achieve the purpose of cleaning the surface and pores of the material. We could infer that this "self-

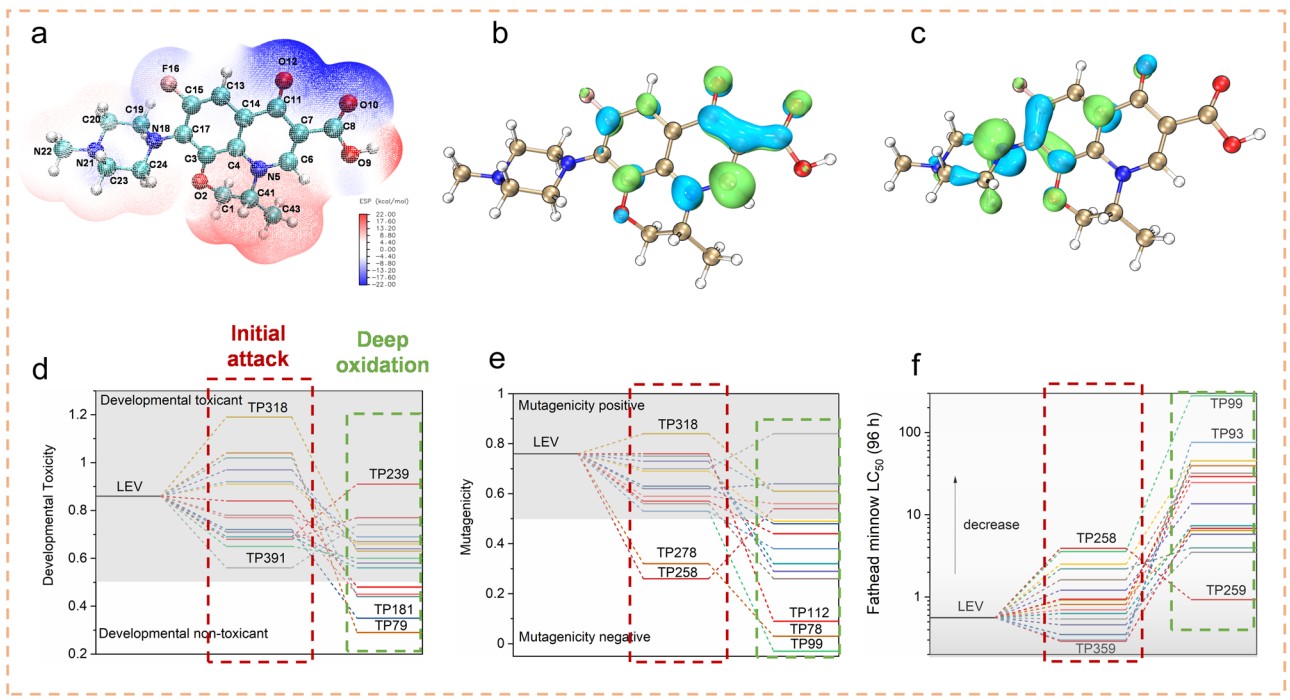

**Fig. 3 | DFT calculations on Fukui index of LEV at the level of B3LYP/6-31 G\*. a** The electrostatic potential (ESP) distribution. **b** HOMO and (**c**) LUMO orbital distribution of LEV molecule. **d**–**f** The toxicity of LEV and its degradation products including developmental toxicity, mutagenicity, and Fathead minnow LC50 (96 h).

purification effect" promotes a faster inner-surface renewal rate, which is one of the important reasons for the excellent degradation performance of BTT-DATP-COFs.

The toxicity of LEV and TPs is also estimated by T.E.S.T.[38,39]. Three parameters, *i.e.* developmental toxicity, mutagenicity and *Fathead minnow* LC$_{50}$ (96 h), are obtained based on the quantitative structure-activity relationship (QSAR) method. For the mutagenicity toxicity (Fig. 3d), LEV is recognized as a high developmental toxicant chemical with a value of 0.86. All the macromolecular intermediates after initial attack show developmental toxicant, especially for TP318 with a developmental toxicant value of 1.19. However, the small-molecule TPs such as TP181 (0.35) and TP79 (0.29) after deep oxidation are detoxified into developmental non-toxicant. Similarly, for the mutagenicity parameter (Fig. 3e), LEV (0.76) and most macromolecular TPs are considered mutagenicity-positive chemicals, and only TP278 (0.32) and TP258 (0.26) are detoxified to mutagenicity negative. The deep oxidation was conducive to detoxification, and most small-molecule TPs are mutagenicity negative, especially for TP99 with a relatively low mutagenicity value of -0.03. For the *Fathead minnow* LC$_{50}$ (Fig. 3f), LEV with a *Fathead minnow* LC$_{50}$ value of 0.56 mg/L is considered "harmful", while most of the macromolecular TPs turn to low toxicity compounds due to the increased LC$_{50}$ values. After deep oxidation, the small-molecule TPs exhibit much higher LC$_{50}$ values, especially for TP 99 and TP93 with the LC$_{50}$ values of 279.42 and 75.82 mg/L, suggesting dramatic reduction of toxicity after degradation reaction.

## Adaptability to complex media of water environmental

The influences of various water chemistry factors on LEV removal were studied, including pH, inorganic ions, and natural organic matter (NOM). In a wide pH range of 5–9 (Fig. 4a and Supplementary Fig. 28), BTT-DATP-COF displayed excellent removal performance with high efficiency of 100% within 120 min. However, there was an obvious inhibit effect at pH 3, which was ascribed to the mutual affiliation of LEV zwitterion and BTT-DATP-COF surface charge (Supplementary Fig. 28b). Four representative inorganic anions (Cl$^-$, SO$_4^{2-}$, NO$_3^-$ and NO$_2^-$) and cations (Na$^+$, K$^+$, Cu$^{2+}$ and Al$^{3+}$) in Supplementary Fig. 29 were evaluated for affecting LEV degradation. Clearly, the removal

performance of LEV by BTT-DATP-COF was slightly deteriorated in the presence of inorganic anions. The multivalent cations (Cu$^{2+}$ and Al$^{3+}$) completely inhibited the degradation of LEV due to the shelter effect of the cations to electronegative BTT-DATP-COF surface (Supplementary Fig. 29b). In a weakly acidic solution, the stronger adsorption of multivalent cations than monovalent ions on electronegative, which consequently hindered the reaction of the LEV molecules with the active site. The addition of natural humic acid (HA), sodium humic acid (HA-Na) and fulvic acid (FD) greatly improved the adsorption capacity. Meanwhile, all the three NOMs slightly reduced the removal efficiency of LEV by BTT-DATP-COF within 120 min, and a negligible effect was observed even with the increase of HA concentration (Supplementary Fig. 30). It contributes to the bridging function of NOM between LEV and BTT-DATP- COF[40], detailed analysis is provided in the Supporting Information (Supplementary Note 4). Eyeing actual application, four typical water bodies were employed as the real substances, being involved in tap water (TW), river water (RW), and lake water (LW). Compared with ultrapure water (UW), the removal of LEV in TW and RW by BTT-DATP-COF was distinctly stimulative. Specifically, a removal efficiency of 100% could be reached in respective 60 min in the TW system (Supplementary Fig. 31). It indicates that the presence of sacrificial species including organic and inorganic matters has no significant influence on the removal process and even promotes it. These observations clearly that BTT-DATP-COF photocatalytic systems have excellent practical application potentials to treat LEV-polluted water.

To validate the universality and practicability of eliminating antibiotics over all the three COFs, the degradation of other antibiotics including chlortetracycline (CTC) tetracycline (TC), ciprofloxacin (CIP), and norfloxacin (NOR) were performed under visible light irradiation in Fig. 4b and Supplementary Fig. 32. The degradation efficiencies of both BTT-DAB-COF and BTT-DADP-COF for CTC and TC reached up to more than 90%, while that of BTT-DATP-COF could almost achieve 100%. The removal efficiency of BTT-DATP-COF to CIP and NOR degradation was more than 85%, which was better than the other two COFs. Besides, BTT-DATP-COF displayed a high degradation kinetic rate compared with BTT-DAB-COF and BTT-DADP-COF in

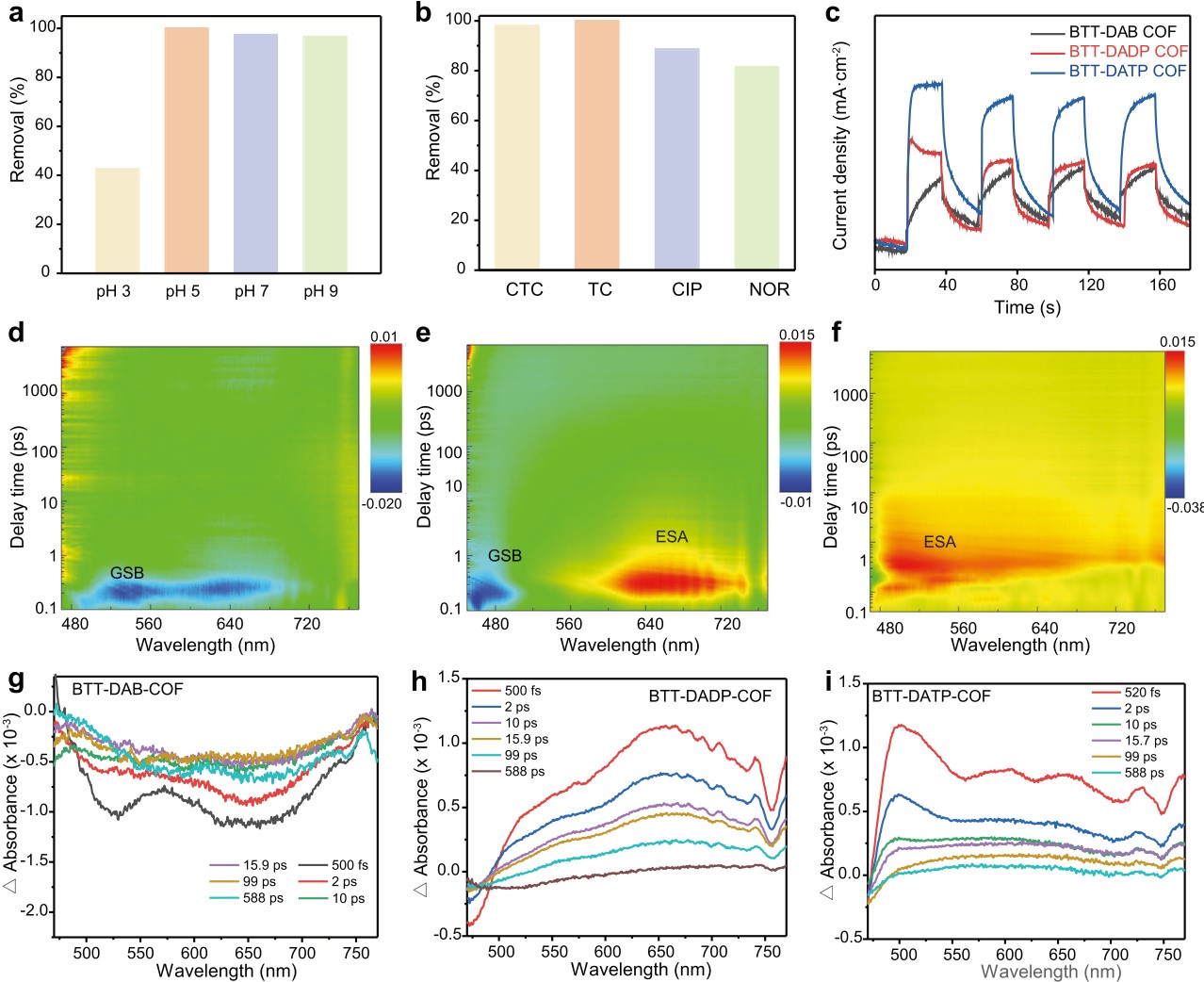

**Fig. 4 | Practicability and carrier dynamics of COFs. a, b** Photocatalytic performance for LEV degradation with different pH, and toward other four typical emerging contaminants for BTT-DATP-COF. **c** Transient photocurrents spectra of three COFs. **d–f** 2D mapping TA spectra of BTT-DAB-COF, BTT-DADP-COF, and BTT-DATP-COF, and (**g–i**) TA spectra signals of three COFs on the fs-ns timescales.

Supplementary Fig. 33, which was 0.02850, 0.03751, 0.01615, and 0.01278 min$^{-1}$ for CTC, TC, CIP and NOR respectively. These results do not only demonstrate the general applicability of eliminating antibiotics by COFs, but also reflect that the BTT-DATP-COF has more accessible channels for mass transportation and has been tightly correlated to inherent photoelectronic properties.

**Originally driving force of LEV degradation by COFs**

The LEV degradation is originally driven by photo-electronic dynamics of COFs, in particular, the separation and recombination of photogenerated carriers (h$^+$–e$^-$) at surface/interface. Steady-state photoluminescence (PL) spectra in Supplementary Fig. 34 further exhibited an obvious PL quenching of BTT-DATP-COF, in contrast with BTT-DAB-COF and BTT-DADP-COF. Femtosecond time-resolved transient absorption (fs-TA) spectroscopy was used to investigate the photoexcited carrier dynamics in these COFs. As shown in Fig. 4d–f, TA spectra with different time delays under pulse excitation were recorded. BTT-DAB-COF showed obvious ground state bleach (GSB) signals at 525 nm and stimulated emission (SE) around 650 nm, and basically recovered within 10 ps (Fig. 4g). The GSB signal was caused by neutral singlet excitons, associated with characteristic simulated emission[41]. In contrast, BTT-DADP-COF and BTT-DATP-COF exhibited extensive excited state absorption (ESA) features that developed on ultrafast

time scales (Fig. 4e, i). The ESA with the center of 500 nm and 650 nm appeared immediately after the photoexcitation of BTT-DATP-COF, attributing to excimer formation (electron excitation). Interestingly, a negative peak can be observed at 750 nm for BTT-DADP-COF and BTT-DATP-COF, which are reasonably interpreted as a photoinduced emission process originating from the interfacial charge transfer (labeled as ICT signal) from DADP/DATP to BTT units[42]. The above phenomena provide direct evidence that the twist effect promotes charge transfer and effectively inhibits electron-hole recombination. Previous studies have shown that the optical density (ΔOD) signal amplitude is proportional to the exciton population[43]. It infers that the number of excitons for BTT-DATP-COF and BTT-DADP is much more than that of BTT-DAB-COF. The fitting dynamics reveal the variation of two decay time constants in Supplementary Fig. 35, and found that the $\tau_1$ and $\tau_2$ of BTT-DAB-COF (0.6 ps) have no change significantly, due to no excited state absorption and only having one component attenuation. The average lifetime of BTT-DATP-COF (75 ps) and BTT-DADP-COF (64 ps) is much longer than BTT-DAB-COF (3.23 ps) due to the generally longer lifetime of excimer formation and decay compared to singlet exciton emission. The results of carrier lifetime are consistent with the crystal sizes of BTT-DAB-COF, BTT-DADP-COF and BTT-DATP-COF, corresponding to respective 16.80, 25.88, and 32.30 nm (Supplementary Note 1). That is, the increase of crystal size,

coupled with a more ordered and conjugated structure, effectively suppresses carrier recombination[44]. Considering DATP with a certain flexibility and twist effect[27], it can be inferred that upon absorbing a certain amount of energy from visible light irradiation, BTT-DATP-COF contribute to a free rotation between DATP and imine bond together at the arm of the framework, which then accelerates carriers migration through the kinetic energy conversion and inhibiting electron-hole recombination. Based on this, we confirmed that the twist effect not only controls the crystallinity and conjugation during the synthesis of COFs, but also is a prerequisite for establishing longitudinal and transverse charge-transfer channels between terphenyl and BTT motif.

Actual surface/interfacial behavior of charge into the enhanced removal performance was obtained by photoelectrochemical (PEC) experiments using a three-electrode system. According to Fig. 4c, the BTT-DATP-COF showed markedly higher photocurrent density than BTT-DAB-COF and BTT-DADP-COF. It was indicated that the BTT-DATP-COF had the upper charge separation capability and the more available surface photogenerated carriers for solid-liquid interfacial reaction[45,46]. Electrochemical impedance spectroscopy (EIS) analysis revealed that the BTT-DATP-COF had a lesser charge transference resistance, resulting in the faster transfer of electrons at the interface (Supplementary Fig. 36 and Supplementary Table 10), especially in a higher frequency. The tendency of photoelectrochemical results is positively correlated with that of LEV removal. Because there are various accumulations of surface charge carriers for participate in the evolution of active substances.

Thermodynamically, the VB potential (2.28 eV) of BTT-DATP-COF is not enough to produce ·OH originating from water ($E_{(H2O/·OH)}$ = 2.37 eV Vs NHE). We speculated that the ·OH in the BTT-DATP-COF system might be derived from hydrogen peroxide ($H_2O_2$) due to the simultaneous presence of $e^-$ and ·$O_2^-$ ($E_{(O2/·O2-)}$ = -0.33 eV Vs NHE according to Supplementary Eq. 10–6. From Supplementary Fig. 37, the $H_2O_2$ was immediately generated under the visible light irradiation, and continued to increase with the prolongation of time. A concentration of 120 μmol/L of $H_2O_2$ was finally detected within 120 min. Besides, hydrophobic BTT-DATP-COF reduces the excess supply of protons ($H^+$), inducing the decomposition of $H_2O_2$ into ·OH instead of water molecules (Supplementary Eqs. (22) and (27)), further enhancing the degradation of pollutants. The evolution pathways of active species in BTT−TPDA−COF system might undergo as Supplementary. Although all of $h^+$, ·$O_2^-$, ·OH, and $^1O_2$ worked in LEV removal, the photoinduced $h^+$ is the most predominant.

## Theoretical calculation and analysis of both COFs and LEV degradation

In order to deeply explain the dynamic difference of LEV degradation by the three COFs, DFT calculations on the interaction of LEV molecule and COFs are performed, mainly involving the adsorption energy and the effect of spin structures related to electronic states, and the selection of COFs periodic structure is shown in Supplementary Fig. 38. The HOMO-LUMO distributions for three COFs are displayed in Supplementary Fig. 39. The HOMO of three COFs is localized at the BTT unit, while the LUMO is concentrated on the DAB, DADP, and DATP segments, respectively. Obviously, the distribution of HOMO and LUMO are more uniform in BTT-DAB-COF and BTT-DADP-BTT, as the degree of conjugation decreases[47]. The partial density of states (PDOS) of COFs shows that a new molecular orbital mainly contributed by electrons from S forms the HOMO, and the other molecular orbital contributed by electrons N constitutes the LUMO (Fig. 5a, b and Supplementary Fig. 40). This type of molecular orbital exhibits slightly lower energetic level than that of the molecular orbital contributed by C. It is still confirmed that the photo-formation of holes pivotal in the LEV degradation is closely related to thiophene units in BTT.

The adsorption energy of LEV for pore-site BTT-DAB-COF, BTT-DADP-COF, and BTT-DATP-COF is -0.38, -0.37 and -3.02 eV in Fig. 5c

and Supplementary Fig. 41, indicating that BTT-DATP-COF has stronger affinity to LEV molecule than BTT-DAB-COF and BTT-DADP-COF. It verifies that the successful creation of the COF surface microstructure has also made it possible to break the contact barrier between the catalyst surface and pollutant. To gain deep insights into the enhanced adsorption of LEV molecules on COFs, the PDOS plots of LEV and COFs are calculated (Fig. 5d, e). The overlap between COFs and LEV bands in the PDOS represents the adsorption strength[48]. The PDOS of BTT-DATP-COF overlaps most with LEV molecule, showing obvious interaction. In addition, the PDOS curve of the BTT-DATP-COF and BTT-DADP-COF shifted toward the left (i.e., lower energies compared with that of BTT-DAB-COF, revealing a more stable adsorption[48]. The oxidation of thiophene units with LEV molecule (purple dashed line region in Fig. 5g) is significantly enhanced in the BTT-DATP-COF with the adsorption energy of −0.452 eV (Fig. 5h). This reveals that the piperazine segments of LEV are more likely to combine with BTT motif, which is then oxidized by holes ($h^+$). It is consistent with the results of the degradation pathway and the calculation of the Fukui index (Supplementary Figs. 26 and Fig. 3d). Furthermore, the adsorption energy at the pore site (−3.02 eV) is much higher than the top site (−0.452 eV), indicating that the LEV molecule is trapped in the pore size. Due to the AA stacking mode, the available top-site adsorption sites are rather limited, and a large quantity of adsorption sites in pore-sites are the main ones for the large amount LEV adsorption by the COFs. This phenomenon explains that the pore confinement effects of BTT-DATP-COF are one of the important means to regulate its catalytic activity.

Although improved adsorption energy can contribute to LEV decomposition, this alone cannot explain the higher kinetics of photocatalytic degradation. The activation mechanism of reactive oxygen species (ROS) is also considered. As shown in Fig. 5i and Supplementary Figs. 42, 43, the $O_2$ molecule can be more easily adsorbed to the DAB, DADP, and DATP function motif (electron-acceptor unit) rather than the BTT units (electron-donor unit). The adsorption energy of BTT-DATP-COF (0.544 eV) was lower than BTT-DAB-COF (0.557 eV) and BTT-DADP-COF (0.584 eV). In this way, the excited electron from BTT unit transferred to the adsorption site (in DATP units) could be more favorable for the generation of ·$O_2^-$. Considering that $H_2O_2$ is a key species involved in LEV degradation, the adsorption model of $H_2O_2$ on the material is also optimized in Supplementary Fig. 43. The excellent adsorption site of $H_2O_2$ is on DATP units (−0.223 eV), followed by BTT (−0.220 eV) unit and finally imine linkage (0.274 eV). Furthermore, the adsorption energy of BTT-DATP-COF (−0.223 eV) was lower than BTT-DAB-COF (−0.184 eV) and BTT-DADP-COF (−0.185 eV). As a consequence, adsorbed $H_2O_2$ onto the DATP units will accept combine with the transferred electron and automatically being decomposed into more hydroxyl radicals in Fig. 5j. Due to the total Gibbs free energy change (ΔG) (Fig. 5g) for the formation of ·OH on DATP is negative, the process can easily occur in thermodynamics. Differential charge density demonstrates that both $O_2$ and $H_2O_2$ continuously extract electrons from the DATP unit, inhibiting the recombination of electrons and holes. Besides, the formation of ROS related to electrons mainly occurs in the DATP region, which is consistent with the results of the distribution for LUMO−HOMO orbital and the PDOS. Electrons are transferred from the BTT unit to the DATP unit, and residue holes in the BTT unit under photoexcitation also can directly react with LEV.

Based on the above analysis, the whole evolutive process of LEV degradation Supplementary Fig. 44) is summarized as follows: (i) COFs differentially adsorbs LEV molecule firstly, and then undergo the separation and migration of photoinduced electrons and holes under light irradiation; (ii) the formed holes on the BTT fragments can directly oxidize the LEV; iii) in cooperated with $O_2$ and $H_2O_2$, the DATP fragment utilize the transferred electrons for generating active radicals including ·$O_2^-$ and ·OH to attack LEV. Spatially independent redox sites

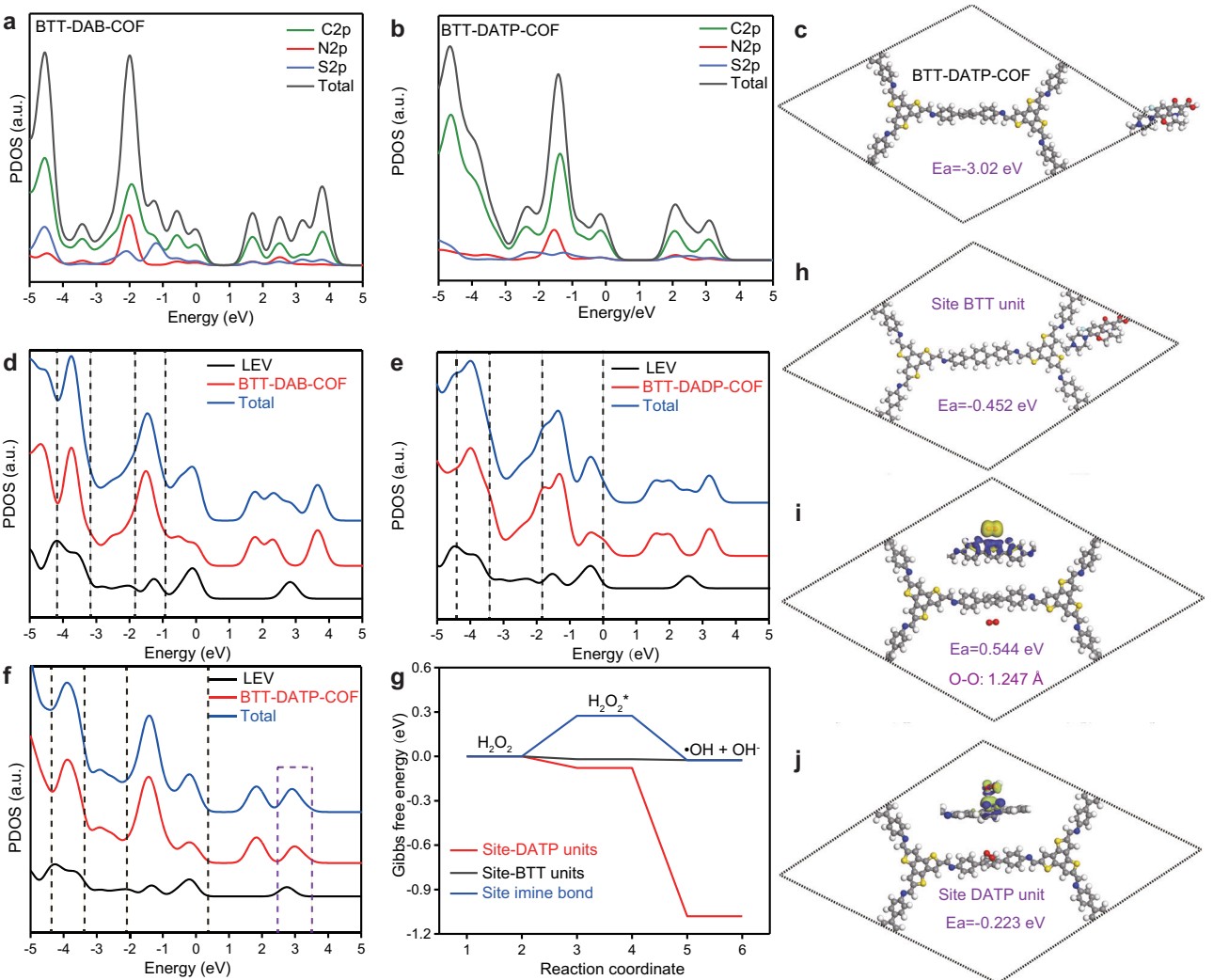

**Fig. 5 | Theoretical calculation both COFs and LEV degradation. a, b** Total density of states (TDOS) and partial density of states (PDOS) BTT-DAB-COF, and BTT-DATP-COF. **c** The favorite adsorption configurations of a LEV molecule adsorbed on BTT-DATP-COF. **d–f** The PDOS of adsorbed LEV molecules: BTT-DAP-COF, BTT-DADP-COF, and BTT-DATP-COF, the dash indicates the overlap areas of adsorbed molecule and COFs. **g** Distribution of dissociation potential energy of $H_2O_2$ on BTT-DATP-COF surface. **h** The favorite adsorption configuration of a LEV molecule adsorbed on BTT fragments. **i, j** The favorite adsorption configurations and the charge difference density of $O_2$ and $H_2O_2$ molecule adsorbed on BTT-DATP-COF.

are formed within the periodic and ordered framework, which inhibits the recombination of electron-hole pairs. The twisty terphenyl unit under photoexcitation provides the sustainable and strong power source for pollutant degradation, the hydrophobic surface structure promotes the interface renewal of BTT-DATP-COF, and the suitable pore assists the transport of small molecules out from the porous COFs material.

## Discussion

In summary, we have rationally explored an integrated interfacial design strategy for constructing an active and robust catalyst for photochemically decomposing contaminants. We designed the photocatalyst by engineering two distinct interfaces to facilitate electron flow and contaminants transport: (1) For the π-electronic interface, the π skeleton was designed to efficiently extend the visible-light redshift and build order π skeletons to promote a seamless electron transfer from the antennae to the catalytic centers; (2) for the mass transport interface, the nanopore of different size were designed to be hydrophobic to break through the adsorption barrier for pollutants on the catalyst, facilitate the contaminants molecules delivery to the reaction centers and improve the self-purification capacity of COFs. The BTT-

DATP-COF has the lowest adsorption energy and could degrade to 100% LEV within 100 min, compared to those obtained with hydrophilic networks or small channels. Meantime, BTT-DATP-COF could efficiently photodegrade LEV in complicated water matrices even in river water, lake water, and top water. Finally, we considered the universality of this series of COFs in the degradation of antibiotic contaminant, and found that the degradation rates of BTT-DATP COF for chlortetracycline (CTC) tetracycline (TC), ciprofloxacin (CIP), and norfloxacin (NOR) all remained above 85%. This study provides an insightful avenue of synergistically regulating electronic structure and interfacial reaction based on organic porous networks toward an energy-saving and cost-reducing way for water purification.

## Methods

1,4-Diaminobenzene (DAB, 98%), 4,4-Diaminodiphenyl (DADP), 4,4-diaminoterphenyl (DATP, 98%) and benzo[1,2-b:3,4-b':5,6-b"]trithio-phene-2,5,8-tricarbaldehyde (Btt, 98%) were supplied Yanshen Technology Co., Ltd (Jilin Chinese). 1,2-dichlorobenzene (o-DCB), mesitylene, and 1,4-dioxane were purchased from Aldric Chemicals. All other reagents and solvents were bought from Sinopharm Chemical Reagent Co., Ltd.

## BTT-DAB-COF

Benzo[1,2-b:3,4-b':5,6-b"] trithiophene-2,5,8-tricarbaldehyde (BTT, 33 mg, 0.1 mmol), 1,4-Diaminobenzene (DAB, 16.2 mg, 0.15 mmol) were put into a 10 ml a pyrex tube, and the dissolved in 3 ml o-dichlorobenzene (o-DCB) and n-butanol mixed solution (V/V = 1:1). After the above mixture was sonicated for 3 min, 0.15 ml of 6 M acetic acid aqueous solution was added, and then sonicated 3 min over again. The tube was degassed by three freeze-pump-thaw- cycles. The tube was sealed off and then heated at 120 °C for 3 days. The powder collected was washed with tetrahydrofuran for three times, and dried at 120 °C under vacuum for 12 h to obtain BTT-DAB-COF.

## BTT-DADP-COF

Benzo[1,2-b:3,4-b':5,6-b"] trithiophene-2,5,8-tricarbaldehyde (BTT, 33 mg, 0.1 mmol), 4,4-Diaminodiphenyl (DADP, 27.7 mg, 0.15 mmol) were put into a 10 ml a pyrex tube, and the dissolved in 3 ml o-dichlorobenzene (o-DCB) and n-butanol mixed solution (V/V = 1:1). After the above mixture was sonicated for 3 min, 0.15 ml of 6 M acetic acid aqueous solution was added, and then sonicated 3 min over again. The tube was degassed by three freeze-pump-thaw- cycles. The tube was sealed off and then heated at 120 °C for 3 days. The powder collected was washed with tetrahydrofuran three times, and dried at 120 °C under vacuum for 12 h to obtain BTT-DADP-COF.

## BTT-DATP-COF

Benzo[1,2-b:3,4-b':5,6-b"] trithiophene-2,5,8-tricarbaldehyde (BTT, 33 mg, 0.1 mmol), 4,4-diaminoterphenyl (DATP, 39 mg, 0.15 mmol) were put into a 10 ml a pyrex tube, and the dissolved in 2 ml mesitylene and 1,4-dioxane mixed solution (V/V = 1:1). After the above mixture was sonicated for 3 min, 0.1 ml of 6 M acetic acid aqueous solution was added, and then sonicated 3 min over again. The tube was degassed by three freeze-pump-thaw- cycles. The tube was sealed off and then heated at 120 °C for 3 days. The powder collected was washed with tetrahydrofuran three times, and dried at 120 °C under vacuum for 12 h to obtain BTT-DATP-COF.

### Reporting summary

Further information on research design is available in the Nature Portfolio Reporting Summary linked to this article.

## Data availability

The data that support the findings of this study are available within the article and its Supplementary Information. Source data are provided with this paper.

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

## Acknowledgements

The authors gratefully acknowledge the financial support provided by the Second Tibetan Plateau Scientific Expedition and Research Program (STEP) (No. 2019QZKK1003 (C.Z)), the National Natural Science Foundation of China (No. 72088101 (X.Y), 22178091 (H.W.), 51708195 (H.W.), 51739004 (X.Y.) and 52270053 (W.L.)), the provincial natural science foundation of Hunan (No. 2023JJ10012 (H.W.)), the National Key Research and Development Program of China (No. 2021YFC1910400 (L.T) and 2021 YFAI202500 (W.L.)), the science and technology innovation Program of Hunan Province (No. 2022RC1120 (H.W.)), the Key Laboratory of Advanced Functional Composites Technology (No. 6142906210508 (X.W.)), and the Fundamental Research Funds for the Central Universities (No. 531118010675 (H.W.)).

## Author contributions

C.Q. performed experiments, analyzed data and writing this paper; Y.Y. and H.W. came up with the original ideal, performed photocatalytic experiments and process data; X.W. help with calculation part; L.C., Z.L., and W.L. conducted the DFT calculations on Fukui index; D.H., D.W., C.Z., L.T., L.L., and X.Y. help with the data interpretations; W.L. and H.W. supervised the project; C.Q, W.L., and H.W. wrote the manuscript; all authors commented on the manuscript.

## Competing interests

The authors declare no competing interests.
