## [Peer Review File · Nature Communications]

Twistedly Hydrophobic Basis with Suitable Aromatic Metrics in Covalent Organic Networks Govern Micropollutant DecontaminationREVIEWER COMMENTS

Reviewer #1 (Remarks to the Author):

Twistedly Hydrophobic Basis with Suitable Aromatic Metrics in Covalent Organic Networks Govern Micropollutant Decontamination

Chencheng Qin, Yi Yang, Xiaodong Wu, Long Chen, Xingzhong Yuan, Lin Tang, Danlian Huang, Dongbo Wang, Lai Lyu, Wen Liu, and Hou Wang

Among the three COFs based photocatalysts described here for water treatment, two of them were already reported by other group (J. Am. Chem. Soc. 2018, 140, 11618–11622) and the other one is just a simple extension of the amine linker of those reported COFs. Although visible light induced photocatalytic degradation of levofloxacin antibiotics ((LVF) using COFs material for water filtration/treatment is relatively new, however, the saturated adsorption capacity (q_m) and cyclic stability of these COFs based photocatalysts is significantly lower compared to other reported adsorbents. The detailed photocatalytic degradation mechanism of the antibiotic (LVF) and removal of their byproducts from the adsorbents (after treatment) is already reported elsewhere. Therefore, in terms of the novelty of this work, I feel the manuscript is a bit short to meet the high criterion of the leading journal of Nature Communication, and would reserve my recommendation until the following critical issues addressed.

The authors should clearly mention earlier report of these COFs and should definitely cite the work (J. Am. Chem. Soc. 2018, 140, 11618–11622). The PXRD peak positions and morphology of the described COFs are significantly different from the earlier reported COFs. They need to justify why they observed different phases of the reported COFs (BTT-DAB-COF and BTT-DADP-COF) although they had followed a similar synthetic method/condition? The pore size of the respective COFs (BTT-DAB-COF, BTT-DADP-COF and BTT-DATP-COF) is expected to be decreased on the following order from the diffraction angles of (100) planes of 2.26° , 2.78° and 3.50° . However, the opposite trend (2.2, 2.7 and 3.2 nm respectively; Fig. 2a) is observed. The authors need to be explained the statement.

Although the author claimed that relatively narrow absorption is observed BTT-DAB-COF (640 nm) and an obvious red-shift absorption edge was observed for BTT- 182 DADP-COF (~ 10 nm) and BTT-DATP-COF (~ 20 nm). However, from the figur2b, the opposite

phenomenon is observed and that seems to be logically acceptable due to more planer DAB linker (torsional angle 28.5) compared to DADP (145.8) and DATP (146.9)

The authors need to provide some basic characterizations (Physical and morphological) of the described COFs after the adsorption of the antibiotics levofloxacin hydrochloride (LEV) Since the adsorption process is a surface phenomenon, the adsorption capacity of the antibiotics is expected to be increased with surface area, however here adsorption process is independent of the BET-surface area values of COFs. The pore size and surface area is also not correlated for these COFs. The authors need to be explained the strange phenomenon” The long-term stability (Chemical Stability & Photostability) of the photocatalysts (COFs) in harsh condition (strong acid and base and boiling water) is also need to be assessed for the application of the water treatment. Moreover, the linkers (1,4-diaminobenzene and its analogs) used for the synthesis of these COFs is known as very toxic to aquatic organisms. “The author claimed that better dispersion of the BTT-DATP-COF (contact angle 136°) compared to the BTT-DAB-COF 224 (60°) and BTT-DADP-COF (122°). However, the materials with lower contact angle is easily dispersed in water is logically correct. On the basis of this point, BTT-DADP-COF (122°) should have better adsorption ability for the antibiotics (LEV) The π - π stacking interaction between COFs and LVF was completely neglected for the adsorption of LVF into COFs

To claim the favorite absorption of ROS-generating species (O₂ and H₂O₂) into the DATP function motif of BTT-DATP-COF, the adsorption interaction of these molecules should also be considered for the remaining COFs

Besides, the high- resolution TEM (HRTEM) images exhibited the interlayer stacking of 3.9, 3.6 and 3.3 Å (BTT-DAB-COF, BTT-DADP-COF and BTT-DATP-COF), which is not correlated to interlayer stacking distances observed from the typical (001) crystal plane of the PXRD pattern.

The authors need to provide a zoom version of EIS spectra at a higher frequency as an inset image. The lower charge transfer resistance of BTT-DATP- COF is not clearly understood from the given figure (Figure S24)

Reviewer #2 (Remarks to the Author):

In this manuscript, the authors prepared three kinds of COFs by modulating different aromatic metrics, and investigated their interfacial behavior in advanced oxidation

decontamination of levofloxacin hydrochloride. Through the optimization of building pieces, a high removal rate for typical micropollutant was achieved by BTT-DATP-COF. The authors also investigated the mechanism for photocatalytic decomposition of COFs toward typical antibiotic by TA and DFT. However, the evidence is not enough to support the statement on the role of aromatic metrics in COF photocatalyst. In addition, the mechanism of photocatalytic decomposition is still not clear, and there are many problems in the manuscript. Thus, I cannot recommend this manuscript for publication in this journal. Some special comments are listed as follows.

1.The authors claim that BTT-DATP has a narrow FWHM compared with that of BTT-DAB-COF and BTT-DADP, which indicates that BTT-DATP shows a higher crystallinity. However, I observed the opposite result from Figure 1b-d. The XRD peak signal of BTT-DATP-COF is relatively weak. In addition, the BET surface area of BTT-DATP-COF is lower than that of other two COFs, suggesting that BTT-DATP exhibits a low crystallinity. As for peak intensity, it is related to the parameters set by the instrument.

2.Whether BTT-DAB-COF and BTT-DADP are new COF materials? If not, please cite the related references.

3.For the pore size (Figure 2a), how to explain the pore distribution at ~ 1.8 nm for all the COFs?

4.As the temperature rises, the weight of BTT-TPDA-COF increased from the TGA curves. Why? Moreover, BTT-DADP-COF begins to decompose at 300 oC. So, the description of the thermal stability for these COFs is nor correct.

5.In Figure S7, the lattice fringe is very clear in the local enlarge images. However, I cannot observe the lattice fringe in HRTEM images. This is very strange. Please provide the clear images.

6.In page 10, in comparison with BTT-DAB-COF, there should be an obvious blue-shift (not red shift as mentioned by the authors) of the optical absorption edge for BTT-DADP-COF and

BTT-DATP-COF. Is this phenomenon consistent with the followed analysis?

7. In page 12, I observe that BTT-DATP-COF shows a slow adsorption kinetic among the COFs studied here, as seen in Figure 2c. The adsorption kinetic (k_2) of BTT-DATP-COF was 0.000426 g.mg/s, less than 60-fold that of DTT-DAB-COF (0.02551 g.mg/s). In addition, the calculated mass-transfer coefficient (k_f) values for the LEV are also inconsistent with the absorption kinetics results. So, this result is not enough to support the conclusion of the manuscript.

8. Why use aqueous HClO₄ to adjust the initial pH value? As we know, HClO₄ is a strong oxidant, which can oxidize and degrade pollutants.

9. Please provide the detection method for the concentrations of F⁻, NO₃-N and DON after reaction.

10. The authors claim that TP186 and TP94 are the main intermediates. It is recommended to use the COF as photocatalyst to degrade TP186 and TP94, respectively. If the final products can be obtained by using this method?

11. In Figure S17, what is the concentration of inorganic anions and cations?

12. The authors studied the influence of HA on the adsorption capacity. Why select HA? Can other aromatic acid promote this process?

13. It is suggested to unify the name of material. The name of COFs in Figures S4, S5, and S21 are not consistent with that in other Figures.

14. In Figure S29, the adsorption energy between BTT and O₂ (0.583 eV) is higher than that of DATP and O₂ (0.544 eV). Why the authors think that the O₂ molecule can be more easily adsorbed to the DATP function motif rather than the BTT units?

15. As reported previously (J. Am. Chem. Soc. 2013, 135, 546–549, J. Am. Chem. Soc. 2018,

47, 16124–16133, and Angew. Chem. Int. Ed. 2013, 52, 13052-13056), the interlayer interaction affects the stacking behavior between layers, thus significantly controlling the crystallinity, porosity, and stability of the COFs. Why BTT-DATP-COF with largest torsional angle has smallest distance of interlayer stacking and best crystallinity?

16. Authors should use the MOTT to verify the conduction bands, and use the UPS analyses to determine the band position in addition to the XPS analyses.

17. Authors claimed that hydrophobic effect can promote the transport of degraded substances out of the pore and has a better self-purification effect, however, the result of water contact angle can only prove the hydrophobicity in external surface, which can not verify hydrophobic property of the pores.

18. The authors claimed that “increase of crystal size with more ordered and conjugated structure is conducive to charge carrier transport via inhibiting carrier recombination”. Why the very different crystal size of the COFs (16.80, 25.88 nm) exhibit similar carrier lifetime (0.3, 0.31 ps)?

19. Authors are strongly suggested to give more evidences to support that the twist effect of terphenyl can inhibit electron-hole recombination.

Reviewer #3 (Remarks to the Author):

This manuscript described the synthesis of three COF materials, which were further used as photocatalysts. Characterizations have been made to analyze the COFs and to evaluate their performance for pollutant photodegradation. Extensive computational studies have been performed to understand the materials as photocatalysts. Although the material design does not stand out from many other works in this field, the fundamental knowledge obtained by the detailed analysis should be of slight interest for the readers of Nature Communications. In addition to some flaws and claims without clear support, the key concerns are about the torsion angles and stacking behavior in the COF models, and the

relationship between hydrophobicity of the materials and their performance. Therefore, I recommend reconsidering this manuscript after a revision, provided that the major findings and conclusions still hold true. The detailed comments and questions are listed as following.

1. The authors highlighted the different torsion angles of the aromatic anilines in the title, introduction (line 97) and Figure 1. However, it is unclear how the torsion angles are obtained. As this manuscript includes extensive computational studies, which use the structural models of the COFs, torsion angles are important and they could affect the computational results.
2. The authors used simulated PXRD patterns to justify the stacking behaviors of the COFs. However, with very few peaks in the experimental PXRD pattern, it is difficult to distinguish different models, e.g., BTT-DADP-COF. In addition, AA inclining and partial AB stacking are other common stacking behavior of 2D COFs. The authors need to add more characterizations such as in-plane TEM images to justify the models.
3. As the authors mentioned, the Scherrer equation can give indication of the size of crystalline domains. Therefore, the authors should also discuss the crystal size of the COFs before draw conclusion of the ordered degree of their π -conjugation system.
4. It is difficult to see the 001 reflection in Figure 1. I suggest to add a zoom-in figure in the SI to show that region.
5. In line 122, the authors described "All of BTT-DAB-COF, BTT-DADP COF and BTT-DATP-COF have high crystallinity with the correspondingly respective (100) plane of 2.26°, 2.78° and 3.50°, respectively." The text is inconsistent with the overall description of the COFs, where BTT-DAB-COF should have the smallest unit cell, and thus the largest 2theta angle of the 100 reflection.
6. In line 247, the authors mentioned "..., appropriate pore size and hydrophobic effect can promote the transport of degraded substances out of the pore, and has a better self-purification effect". It is clear that large pore size can facilitate diffusion, but the authors did not discuss what "self-purification effect" means, and how hydrophobicity can promote such effect, and how the effect is related to the performance of LEV degradation.
7. To highlight the role of the COFs for the photodegradation of LEV, the authors need to show an experiment without using the COF materials as a reference.
8. As photocatalysts, how much quantum efficiency do the COFs provide?

9. The authors should provide error bars for all curves in Figure 2e.

10. As framework materials, what are the local structural models chosen by the authors for the calculation?

Reviewer #1 (Remarks to the Author):

Twistedly Hydrophobic Basis with Suitable Aromatic Metrics in Covalent Organic Networks Govern Micropollutant Decontamination

Chencheng Qin, Yi Yang, Xiaodong Wu, Long Chen, Xingzhong Yuan, Lin Tang, Danlian Huang, Dongbo Wang, Lai Lyu, Wen Liu, and Hou Wang

Among the three COFs based photocatalysts described here for water treatment, two of them were already reported by other group (J. Am. Chem. Soc. 2018, 140, 11618–11622) and the other one is just a simple extension of the amine linker of those reported COFs. Although visible light induced photocatalytic degradation of levofloxacin antibiotics ((LVF) using COFs material for water filtration/treatment is relatively new, however, the saturated adsorption capacity (qm) and cyclic stability of these COFs based photocatalysts is significantly lower compared to other reported adsorbents. The detailed photocatalytic degradation mechanism of the antibiotic (LVF) and removal of their byproducts from the adsorbents (after treatment) is already reported elsewhere. Therefore, in terms of the novelty of this work, I feel the manuscript is a bit short to meet the high criterion of the leading journal of Nature Communication, and would reserve my recommendation until the following critical issues addressed.

Response: Thank you very much for providing valuable feedback and suggestions on our paper. According to the insightful concerns of the reviewer, further elaborations on our innovative points are provided for this study:

To enable a carbon-neutral industrial society, utilizing sustainable and clean solar-energy photocatalysis to solve water crises caused by emerging contaminations has triggered broadened interest. The fact that how to achieve efficiently photochemical decomposition of micropollutants while reducing energy consumption in water treatment has been long ignored, but still a great challenge. Quite limited strategy has been proposed for simultaneous regulation of electronic structure and interfacial reaction from the views of functional basis in solid-liquid interface. In this work, we successfully synthesized three kinds of imine-linked COFs, termed as BTT-DAB-COF, BTT-DADP-COF and BTT-DATP-COF. By altering the functional motif, a subtle structural variation for COFs is realized-especially in the interface, molecular ordering and organic building blocks, which improves the intermediates selectivity and micropollutant decontamination from water. This allows the entire treatment system to move in the direction of energy savings and cost reduction. Several highlights enlightening in a broad area have been shown as follows:

1. A concept of “twistedly hydrophobic basis with suitable aromatic metrics” in periodic frameworks was unveiled for governing COFs synthesis, photoelectrochemical properties and photocatalytic performance. In details, the twist effect does not only control the crystallinity and conjugation during the synthesis of

COFs, but also is a prerequisite for establishing longitudinal and transverse charge-transfer channels between terphenyl and BTT motif.

2. The twist effect of terphenyl unit would pull imine bond to rotate freely under photoexcitation, improve the electronic structure of COFs via offering a higher π -conjugation and thus prolonging electron-hole lifetime. The interfacial charge transfer (labelled as ICT signal) from DADP/DATP to BTT units observed by the fs-TA provide direct evidence for this concept.

3. Both favorable surface characteristic and pore size solve the limitation of mass transfer and reactive sites exposure, improving the inner-surface renewal and breaking through adsorption energy barrier. Based on the relationship between the hydrophilic by-products and a hydrophobic inner surface of materials, it is proposed that the “self-purification effect” promotes a faster inner-surface renewal rate, which is one of the important reasons for the excellent degradation performance of BTT-DATP-COFs.

4. The surface adsorption energy (-3.02 eV) of the novel BTT-DATP-COF for micropollutants is greater 8~ times than other COFs. The capacity of removing micropollutants for BTT-DATP-COF is 6 and 8 times that of BTT-DADP-COF and BTT-DAB-COF.

5. The hole-driven micropollutant oxidation at the interface of BTT fragments to occur, accompanying with electron-mediated oxygen reduction on terphenyl motif to active radicals, endowing it facilitate the balanced extraction of holes and electrons.

1. The authors should clearly mention earlier report of these COFs and should definitely cite the work (*J. Am. Chem. Soc.* 2018, 140, 11618–11622). The PXRD peak positions and morphology of the described COFs are significantly different from the earlier reported COFs. They need to justify why they observed different phases of the reported COFs (BTT-DAB-COF and BTT-DADP-COF) although they had followed a similar synthetic method/condition?

Response: Thanks for the reviewer’s issue. The mentioned literature has been correctly cited as **Ref. 24** of our revised manuscript.

The synthetic conditions, including the purity of the organic monomer, temperature, catalyst concentration, vacuum degree and so on, have a significant impact on the crystallinity and morphology of COF materials (*J. Am. Chem. Soc.*, 2022, 144, 14, 6583–6593). In this work, we present a comparison of the PXRD peaks and morphologies of two COFs with the previous report (*J. Am. Chem. Soc.*, 2018, 140, 11618–11622) as follows:

Firstly, we meticulously re-evaluated the raw PXRD data. In our study, the diffraction angles of BTT-DAB-COF and BTT-DADP-COF at the (100) plane were found to be 3.50° and 2.78°, respectively. In contrast, the peaks of BTT-DAB-COF and BTT-DADP-COF in the previous work were likely around 3.50° and 2.80°, respectively (*J. Am. Chem. Soc.*, 2018, 140, 11618–11622). The main peak positions almost coincide, as depicted in **Fig. R1**. To provide better clarity, we have included a revised description of the main peaks of COFs on **Page 6** of our revised manuscript.

Fig. R1 Comparison of positions of XRD diffraction peaks (left is this work, right is already reported work)

On the other hand, in terms of morphology, we photographed the high-resolution TEM of BTT-DAB-COF and compared it with the existing literature (only TEM images of BTT-DAB-COF were provided in *J. Am. Chem. Soc.*, 2018, 140, 11618–11622). It is evident that the nanosheet structure is clearly visible under the scale bar of 50 nm. The overall morphology in both works exhibits a similar trend, as illustrated in **Fig. R2**.

Fig. R2 Comparison of BTT-DAB-COF morphology (left is this work, right is already reported work)

2. The pore size of the respective COFs (BTT-DAB-COF, BTT-DADP-COF and BTT-DATP-COF) is expected to be decreased on the following order from the diffraction angles of (100) planes of 2.26°, 2.78° and 3.50°. However, the opposite trend (2.2, 2.7 and 3.2 nm respectively; Fig. 2a) is observed. The authors need to be explained the statement.

Response: Thanks for the reviewer's issue. We carefully re-checked the original data

and uploaded the PXRD data accompanying by the revised manuscript for reference. The description was corrected as follows:

Revised at Page 6 of Revised Manuscript, “All of BTT-DAB-COF, BTT-DADP-COF and BTT-DATP-COF have high crystallinity with the correspondingly respective (100) plane of 3.50° , 2.78° and 2.26° , respectively (**Fig. S2**).

Revised at Page 8-9 of Revised Manuscript, “The pore size of BTT-DAB-COF, BTT-DADP-COF and BTT-DATP-COF corresponded to 2.2, 2.7 and 3.2 nm in **Fig. 2a**, respectively, which aligns with the decreasing trend observed for the diffraction angles at the (100) plane.”

Supplementary Figure 2. XRD pattern of three COFs (the zoom-in figure shows the location of the broad peak of COF)

3. Although the author claimed that relatively narrow absorption is observed BTT-DAB-COF (640 nm) and an obvious red-shift absorption edge was observed for BTT-182 DADP-COF (~ 10 nm) and BTT-DATP-COF (~ 20 nm). However, from the **Fig. 2b**, the opposite phenomenon is observed and that seems to be logically acceptable due to more planer DAB linker (torsional angle 28.5°) compared to DADP (145.8°) and DATP (146.9°).

Response: The reviewer’s suggestions are really appreciated. In order to clarify the light absorption of COFs, the original data of UV-visible diffuse reflectance spectrum (UV-vis DRS) was further normalized for comparison. The results of UV-vis DRS were reanalyzed as follows:

By the UV-Visible diffuse reflectance spectrum (UV-Vis DRS) in **Fig. 2b**, BTT-DAB-COF exhibited relatively narrow optical absorbance with an absorption onset at 648 nm, a red-shift of the optical absorption edge was observed for BTT-DADP-COF (~ 10 nm) and BTT-DATP-COF (~ 20 nm). The corresponding energy band gaps (E_g) decreased. This suggests that terphenyl units enhance the conjugation effect across the plane (along the c plane of z axis) of the imine-linked BTT-based COFs, resulting in a narrower band gap, which is favorable for electronic transitions (*Adv. Mater.*, 2022, 34, 2203139). On the one hand, a donor-acceptor (D-A) torsional spring between BTT and aromatic aniline (DAB, DADP or DATP) undergoes force-induced planarization during uniaxial elongation leading to red-shifted absorption (*Nat. Commun.*, 2021, 12, 4243). On the other hand, reversible bond formation and structural self-healing have a central role in achieving long-range crystalline order for COFs (*Nature.*, 2022, 604, 72-79). More reversible chemistry can improve crystallinity (*Nat. Chem.*, 2013, 5, 830-834). In the presence of reversible imine linkage, the more torsion of the DATP building block

than that of DAB and DADP probably strengthen the structural self-healing of COFs during the dynamic polymerization between BTT and aromatic aniline, thus achieving more long-ranged order COFs. This is consistent with the result that the order of crystallinity for BTT-DAB-COF, BTT-DADP-COF and BTT-DATP-COF.

Revised at Page 10-11 of Revised Manuscript, “By the UV-Visible diffuse reflectance spectrum (UV-Vis DRS) in **Fig. 2b**, BTT-DAB-COF exhibited relatively narrow optical absorbance with an absorption onset at 648 nm, a red-shift of the optical absorption edge was observed for BTT-DADP-COF (~10 nm) and BTT-DATP-COF (~20 nm). The corresponding energy band gaps (E_g) decreased from 2.21, to 2.18, and 2.15 eV (**Fig. S10**). This suggests that terphenyl units enhance conjugation effect across the plane (along the c plane of z axis) of the imine-linked BTT-based COFs, resulting in a narrower band gap, which is favorable for electronic transitions³². A donor-acceptor (D-A) torsional spring between BTT and aromatic aniline (DAB, DADP or DATP) undergoes force-induced planarization during uniaxial elongation, leading to red-shifted absorption³³. This is consistent with the result that BTT-DATP-COF has the highest crystallinity, as well as the result of (001) plane shift.”

Fig. 2a UV-vis DRS spectra and the Kubelka–Munk-transformed reflectance spectra.

4. The authors need to provide some basic characterizations (Physical and morphological) of the described COFs after the adsorption of the antibiotics levofloxacin hydrochloride (LEV).

Response: Thanks for the valuable suggestion of the reviewer. We have conducted additional characterizations of the three COFs before and after levofloxacin hydrochloride (LEV) adsorption, including SEM images for morphologies characterization, N₂ adsorption-desorption isotherm for porous structure analysis and Fourier transform-infrared spectroscopy (FT-IR) spectra for functional groups and bond analysis. Detailed descriptions have been offered in the revised manuscript.

Added in Page 20-21 of Revised Supporting Information, “As shown in **Fig. S16**, the overall morphology of the three COFs did not significantly change before and after adsorption. The specific surface area and porous structure were further analyzed, as displayed in **Fig. S17**. After LEV adsorption, the specific surface area of BTT-DAB-COF, BTT-DADP-COF and BTT-DATP-COF decreased by 48%, 36% and 37%. The pore volume also decreased by 38%, 25% and 21%. The above results revealed that the LEV were adsorbed effectively on the outer surface of COFs and even entre into

internal pore channels. Although BTT-DATP-COF had the highest saturated adsorption capacity for pollutants, it showed a much smaller reduction in specific surface area and pore volume than that of BTT-DAB-COF, demonstrating that an appropriate pore structure can not only facilitate the adsorption of pollutants but also their subsequent diffusion and transport.

As shown in **Fig. S18**, Fourier transform-infrared spectroscopy (FT-IR) spectra of three COFs before and after the adsorption of LEV display similar characteristic peak, indicating the adsorption behavior of LEV onto COFs. The peak located at 3000-3500 cm^{-1} and 1000-1800 cm^{-1} were ascribed to the vibration of O-H deformation vibrations. With LEV adsorbed onto COFs, the peaks of O-H bond became weaker and were shift to higher frequency region. These changes could be attributed to the presence of π - π stacking effect between LEV and COFs, resulting in changes in bond type and the bonding state, and thereby decreasing the variation of dipole moment (*J. Hazard. Mater.*, 2022, 421, 126680; *Appl. Catal. B-Environ.*, 2022, 310, 121298).”

Supplementary Figure 16. SEM of pure COFs (a-c) and COFs adsorbed with antibiotics levofloxacin hydrochloride (d-f).

Supplementary Figure 17. N₂ adsorption-desorption isotherms of COFs before and after the LEV adsorption.

Supplementary Figure 18. FT-IR spectra of COFs before and after LEV adsorption

5. Since the adsorption process is a surface phenomenon, the adsorption capacity of the

antibiotics is expected to be increased with surface area, however here adoption process is independent of the BET-surface area values of COFs. The pore size and surface area is also not correlated for these COFs. The authors need to be explained the strange phenomenon”

Response: Thanks for the reviewer’s issue. We agree that the adsorption capacity of pollutants is generally related to the surface area of adsorbents. The adsorption capacity of pollutants on the surface of materials is also determined by a variety of surface characteristics, including porosity, pore size, pore volume, surface hydrophilic/hydrophobic, amount of functional groups, *etc.* Therefore, it's not usually determined by one factor, but a integrate consequences (*Nat. Mater.*, 2022, 21, 689-695; *Environ. Sci. Technol.*, 2021, 55, 5371-5381; *Nat. Commun.*, 2019, 10, 3861; *Nat. Commun.*, 2018, 9, 187).

In this work, we confirmed that LEV are adsorbed effectively on the outer surface of COFs and even enter into the internal pore channel. The pore size structure also plays a role in the adsorption of pollutants. Previous research revealed that the pore size was positively correlated with the adsorption of benzene, but not with the BET-surface area values (*Nat. Mater.*, 2022, 21, 689-695). Moreover, hydrophobic carriers have enrichment property for organic contaminants (*Environ. Sci. Technol.*, 2022, 56, 4, 2665–2676). The adsorption capacity order of BTT-DAB-COF < BTT-DADP-COF < BTT-DATP-COF may be due to the π – π interaction with the benzene ring. Therefore, it can be found that the adsorption capacity of the three COFs for organic pollutants is primarily related to the integrity of pore size and π – π interaction.

In general, the size of specific surface area (SSA) depends on the size of micropore volume, and has no obvious relationship with the size of mesoporous volume (*Environ. Sci. Technol.*, 2021, 55, 5371-5381; *ACS Appl. Mater. Interfaces.*, 2018, 10, 30265–30272). In this work, the micropore volume of BTT-DAB-COF, BTT-DADP-COF and BTT-DATP-COF were 1.05, 0.85 and 0.37 cm³/g, respectively, and this trend was positively correlated with its BET value. Besides, the average pore size of BTT-DAB-COF, BTT-DADP-COF and BTT-DATP-COF (2.2, 2.7 and 3.2 nm, respectively) was inversely proportional to the BET value.

We further highlight the key parameters of COFs related to adsorption capacity in the revised manuscript as:

Added in Page 13 of Revised Manuscript, “Obviously, all the results show that the adsorption capacity is positively correlated with the pore size, independent of the BET-surface area values of COFs.”

6. The long-term stability (Chemical Stability & Photostability) of the photocatalysts (COFs) in harsh condition (strong acid and base and boiling water) is also need to be assessed for the application of the water treatment. Moreover, the linkers (1,4-diaminobenzene and its analogs) used for the synthesis of these COFs is known as very toxic to aquatic organisms.

Response: Many thanks to the reviewer. Followed the reviewer’s suggestion, we assessed the long-term stability of the three COFs in harsh conditions (strong acid and base and boiling water), and we carefully evaluated the dissolution of organic ligands

including DAB, DADP, DATP and BTT in these conditions. The results indicated the good stability of developed COFs in this work.

Added in Page 24-25 of Revised Supporting Information, “Excepting for BTT-DAB-COF, both BTT-DADP-COF and BTT-DATP-COF can maintain good crystallinity after light (**Fig. S23**) in normal wastewater. Under the conditions of strong acid, strong base and boiling water, the crystallinity of the material greatly decreases or even disappears. Usually, the pH range of actual sewage is around 5-9 (*Bioresour. Technol.*, 2022, 347, 126423; *Sci. Rep.*, 2020, 10, 21027), there is no change removing pollutant for of BTT-DATP-COF under such conditions (pH=5-9). Therefore, this does not affect the application of materials in water pollution control.”

To determine the chemical stability and photostability of the three developed COFs, the possible dissolved organic ligands in harsh conditions were detected on an ultrahigh performance liquid chromatography-mass/mass (UPLC-MS/MS). Specifically, 10 mg of COF (BTT-DATP-COF, BTT-DADP-COF or BTT-DAB-COF) was dispersed in solution (with a total volume of 100 mL) under the following three harsh conditions: strong acid condition (pH = 2, 25 °C), strong alkaline condition (pH = 12, 25 °C) and boiling water condition (pH = 7, 80 °C), respectively. After 10 h's reaction under light irradiation, each 1 mL sample was taken from the solution and filtered through a 0.22 μm nylon filter membrane. The ligands in the filtrate were detected on an ultrahigh performance liquid chromatography-mass/mass (UPLC-MS/MS, Dionex UltiMate 3000 Series; MS, Thermo Scientific, USA) equipped with a Zorbax RX-C18 column. The column temperature was set as 30 °C and a sample volume of 5 μL was injected in an electrospray ionization positive mode (ESI+). The mobile phase was a mixture of chromatography-grade water with 0.1% formic acid and methanol at a flow rate of 0.2 mL min⁻¹. The extracted ion chromatogram (EIC) spectrum of the ligands (DAB, m/z = 109.06072; DADP, m/z = 185.10732; DATP, m/z = 261.13862 and BTT, m/z = 330.95518) are shown in **Fig. S24**. No signals and chromatographic peaks can be observed in all the EICs at different harsh conditions, directly indicating that the ligands would not dissociate and dissolve into solution.

In addition, after reaction for the three COFs, each 15 mL sample was taken out for the test of the total organic carbon (TOC), which was measured by a TOC meter (Shimadzu, TOC-L CPH, TNM-1, Japan). The TOC values of the samples for all the three COFs were undetectable, *i.e.*, lower than the limit of detection (LOD, 0.1 mg/L). This result further demonstrated the chemical stability and photostability of the COFs (**in Supplementary Note 3**).”

Added in Page 14-15 of Revised Manuscript, “After enduring four cycles, the removal efficiency of LEV by BTT-DATP-COF remained above 85% (**Fig. S22**), and contained a similar XRD pattern (**Fig. S23**). The developed three COFs showed high chemical stability and photostability even in harsh conditions (strong acid, strong alkaline and water with high temperature), as no organic ligands would dissolve into solution during a long-term test based on LC-MS/MS analysis and total organic carbon (TOC) change (**Fig. S24**).”

Supplementary Figure 23. PXRD pattern spectra of COFs before and after treating with light, boiling water, NaOH and HCl for 10 h.

Supplementary Figure 24. The extracted ion chromatogram (EIC) of DAB ($m/z = 109.06072$) (a), DADP ($m/z = 185.10732$) (b), DATP ($m/z = 261.13862$) (c) and BTT ($m/z = 330.95518$) (d) for COFs in stability test.

7. “The author claimed that better dispersion of the BTT-DATP-COF (contact angle 136°) compared to the BTT-DAB-COF $224 (60^\circ)$ and BTT-DADP-COF (122°). However, the materials with lower contact angle is easily dispersed in water is logically correct. On the basis of this point, BTT-DADP-COF (122°) should have better adsorption ability for the antibiotics (LEV)

Response: Thanks for the concerns of the reviewer. In the original version, we claimed that “The water contact angle of BTT-DAB-COF (60°) was much smaller than that of BTT-DADP-COF (122°) and BTT-DATP-COF (136°), indicating better dispersion into water in a short time.” According to the reviewer’s suggestion, we have firstly corrected our description.

Revised at Page 13 of Manuscript, “The water contact angle of BTT-DATP-COF (136°) was much larger than that of BTT-DADP-COF (122°) and BTT-DAB-COF (60°), indicating the BTT-DATP-COF has difficult dispersion into water in a short time due to the hydrophobic effect.”

The adsorption capacity of pollutants on the surface of materials is determined by a variety of surface characteristics, including surface area, porosity, pore size, surface hydrophilic/hydrophobic, amount of functional groups, *etc.* In this work, the adsorption capacity of LEV on COFs determined by adsorption kinetics and adsorption isotherms

followed the order of BTT-DAB-COF < BTT-DADP-COF < BTT-DATP-COF. Specifically, the saturated adsorption capacity (q_m) of BTT-DATP-COF calculated according to Langmuir isotherm model was 86.65 mg/g, which was much higher than that of BTT-DADP-COF (32.30 mg/g) and BTT-DAB-COF (29.99 mg/g). Obviously, the adsorption capacity was positively correlated with the pore size, independent of the BET-surface area of COFs. Although BTT-DATP-COF had the largest adsorption capacity, the smallest adsorption kinetic coefficient is obtained. Based on the specific phenomenon, the wetting property of three COFs was further conducted. The water contact angle of BTT-DATP-COF (136°) was much larger than that of BTT-DADP-COF (122°) and BTT-DAB-COF (60°), indicating the BTT-DATP-COF exhibited difficult dispersion into water in a short time due to the hydrophobic effect. Due to the steric effect, the small pore size of BTT-DAB-COF would prevent LEV molecules from entering the channels to obtain a low accumulated capacity, and quickly reaching the adsorption equilibrium onto the external surface. Moreover, hydrophobic carriers have enrichment property for organic contaminants (*Environ. Sci. Technol.*, 2022, 56, 4, 2665–2676). The adsorption capacity order of BTT-DAB-COF < BTT-DADP-COF < BTT-DATP-COF may also be due to the π - π interaction with the benzene ring. Therefore, it can be found that the adsorption capacity of the three COFs for organic pollutants is primarily related to the integrity of pore size and π - π interaction.

8. The π - π stacking interaction between COFs and LEV was completely neglected for the adsorption of LEV into COFs.

Response: In line with the reviewer's suggestions, the π - π stacking interaction between COFs and LEV was analyzed by FT-IR and DFT calculation.

Added in Page 14 of Revised Manuscript: "Furthermore, the physical properties and morphology of COFs after LEV adsorption were investigated in Fig. S15-18. It is indicated that the π - π stacking interaction between COFs and LEV shows another driving force for the adsorption of LEV onto COFs."

Added in Page 20-21 of Revised Supporting Information: "As shown in Fig S18, Fourier transform infrared spectroscopy (FT-IR) of the three COFs before and after the adsorption of LEV display similar characteristic peak, indicating the physical adsorption behavior of LEV onto COFs. The peak located at 3000-3500 cm^{-1} and 1000-1800 cm^{-1} could be ascribed to the vibration of O-H deformation vibrations. In the presence of LEV onto COFs, the peaks of O-H bond became weaker and were shift to higher frequency region. These changes could be attributed to the presence of π - π stacking effect between LEV and COFs, resulting in changes in bond type and the bonding state and thereby decreasing the variation of dipole moment (*J. Hazard. Mater.*, 2022, 421, 126680; *Appl. Catal. B-Environ.*, 2022, 310, 121298). Besides, the distance between the central DATP unit of BTT-DATP-COF and the benzene ring of LEV molecules is 3.7Å, indicating the presence of the π - π stacking interaction (Fig. S15)."

Supplementary Figure 18. FT-IR spectra of COFs loaded with antibiotics levofloxacin hydrochloride.

Supplementary Figure 15. Selected fragments highlighting the π - π stacking interaction between BTT-DATP-COF and antibiotics levofloxacin hydrochloride.

9. To claim the favorite absorption of ROS-generating species (O_2 and H_2O_2) into the

DATP function motif of BTT-DATP-COF, the adsorption interaction of these molecules should also be considered for the remaining COFs.

Response: In line with the reviewer's suggestion, the favorite adsorption of ROS-generating species (O_2 and H_2O_2) into the DAB (DADP or DATP) function motif of BTT-DAB-COF (BTT-DADP-COF or BTT-DATP-COF) were respectively analyzed in our revised manuscript.

Added in Page 29 of Revised Manuscript, "As shown in **Fig. S43-44**, the O_2 molecule can be more easily adsorbed to the DAB, DADP and DATP function motif (electron-acceptor unit) rather than the BTT units (electron-donor unit). The adsorption energy of BTT-DATP-COF (0.544 eV) was lower than that of BTT-DAB-COF (0.557 eV) and BTT-DADP-COF (0.584 eV). In this way, the excited electron transferred from BTT unit to the adsorption site (in DATP units) could be more favorable reaction with O_2 molecule for the generation of $\cdot O_2^-$. Considering that H_2O_2 is a key species involved in LEV degradation, the adsorption models of H_2O_2 on the COFs material are also optimized in **Fig. S44**. The optimal adsorption site of H_2O_2 is on DATP units (-0.223 eV), followed by BTT (-0.220 eV) unit and finally imine linkage (0.274 eV). The adsorption energy of H_2O_2 onto BTT-DATP-COF (-0.223 eV) was lower than that of BTT-DAB-COF (-0.184 eV) and BTT-DADP-COF (-0.185 eV). As a consequence, adsorbed H_2O_2 onto the DATP units is inclined to combine with the transferred electron, and automatically being decomposed into more hydroxyl radical."

Supplementary Figure 43. The favorite adsorption configurations of O_2 and H_2O_2 molecule adsorbed on BTT-DAB-COF, BTT-DADP-COF and BTT-DATP-COF.

Supplementary Figure 44. The favorite adsorption configurations of O₂ and H₂O₂ molecule adsorbed on BTT-DATP-COF.

10. Besides, the high-resolution TEM (HRTEM) images exhibited the interlayer stacking of 3.9, 3.6 and 3.3 Å (BTT-DAB-COF, BTT-DADP-COF and BTT-DATP-COF), which is not correlated to interlayer stacking distances observed from the typical (001) crystal plane of the PXRD pattern.

Response: Thanks for the reviewer's issue. An amplifying area of PXRD was provided in the Page 11 of revised Supporting Information.

As shown in Fig. S2, the wide peak of PXRD at 25.21°, 25.96° and 26.44° can be assigned to the (001) plane generated by π - π stacking of 2D layers for the three COFs. The shift toward higher degree for BTT-DATP-COF indicates a decreased distance of interlayer stacking and better degree of longitudinal conjugation. Meanwhile, the high-resolution TEM (HRTEM) images exhibited the interlayer stacking of 3.9, 3.6 and 3.3 Å along the *c* spacing between (001) plane for BTT-DAB-COF, BTT-DADP-COF and BTT-DATP-COF, respectively (*J. Am. Chem. Soc.*, 2020, 142, 4862–4871; *J. Am. Chem. Soc.*, 2016, 138, 9767–9770). Therefore, the trend of wide peak value from PXRD pattern is negatively correlated to the interlayer stacking exhibited by the high-resolution TEM (HRTEM) images, reflecting the (001) plane.”

Supplementary Figure 2. XRD pattern of three COFs (the illustration shows the location of the broad peak of COF)

11. The authors need to provide a zoom version of EIS spectra at a higher frequency as an inset image. The lower charge transfer resistance of BTT-DATP-COF is not clearly understood from the given figure (Figure S24).

Response: According to the reviewer's suggestions, a figure higher frequency and lower charge transfer resistance of COFs was provided.

Revised at Page 25 of Revised Manuscript: "Electrochemical impedance spectroscopy (EIS) analysis revealed that the BTT-DATP-COF had a lesser charge transference resistance, resulting in the faster transfer of electrons at the interface (**Fig. S37 and Table S9**), especially in a higher frequency."

Supplementary Table 9. Summary of the EIS fitting results of impedance data for the three COFs.

Materials	Parameters ^a				
	R_s (Ω)	R_{sc} (Ω)	R_{CT} (Ω)	C_1 (F)	C_2 (F)
BTT-DAB-COF	408.20	6093.00	1.23×10^6	4.12×10^{-8}	2.02×10^{-7}
BTT-DADP-COF	11.19	44.59	4.88×10^5	5.50×10^{-9}	2.25×10^{-5}
BTT-DATP-COF	11.30	45.88	3.52×10^5	5.50×10^{-9}	2.20×10^{-5}

^a R_s is internal resistance; R_{sc} is interfacial resistance; R_{CT} is charge-transfer resistance; C_1 and C_2 are Constant Phase Element (CPE), which are ideal capacitance in analog circuits.

Supplementary Figure 37. EIS spectra of the three COFs. (the illustration of the enlarged diagram of material internal resistance and the simulated circuit diagram).

Reviewer #2 (Remarks to the Author):

In this manuscript, the authors prepared three kinds of COFs by modulating different aromatic metrics, and investigated their interfacial behavior in advanced oxidation decontamination of levofloxacin hydrochloride. Through the optimization of building pieces, a high removal rate for typical micropollutant was achieved by BTT-DATP-COF. The authors also investigated the mechanism for photocatalytic decomposition of COFs toward typical antibiotic by TA and DFT. However, the evidence is not enough to support the statement on the role of aromatic metrics in COF photocatalyst. In addition, the mechanism of photocatalytic decomposition is still not clear, and there are many problems in the manuscript. Thus, I cannot recommend this manuscript for publication in this journal. Some special comments are listed as follows.

Response: Thank you very much for providing valuable feedback and suggestions on our paper. We have described in detail the role of aromatic in COF photocatalyst and the mechanism of photocatalytic decomposition in submitted manuscript. Therefore, we are more than willing to provide further explanation and elaboration on the two mentioned points.

1. For evidences on aromatic metrics of COF photocatalyst: **i)** The building blocks with suitable aromatic metric have different torsion angles relative to the amino component. Although the triphenyl (DATP) has the greatest torsion angle, it is a flexible monomer that has been shown to be more efficient at stacking during polymer reactions (*J. Am. Chem. Soc.*, 2022, 144, 34, 15581–15594). Reversible bond formation and structural self-healing play a central role in achieving long-range crystalline order for COFs (*Nature.*, 2022, 604, 72-79). In the presence of reversible imine linkage, the greater torsion of the DATP building block compared to DAB and DADP likely strengthen the structural self-healing of COFs during dynamic polymerization between BTT and aromatic aniline, thus achieving more long-ranged order COFs. This is consistent with the observed crystallinity order of BTT-DAB-COF < BTT-DADP-COF < BTT-DATP-COF. Our work verifies that the difference in torsion angle of linkers influences the crystallinity, conjugation, and long-range order of COFs. Furthermore, the ordered π skeletons and highest crystallinity would affect the UV-Visible diffuse reflectance spectrum (UV-Vis DRS), leading to red-shifted absorption due to a donor-acceptor (D-A) torsional spring between BTT and aromatic aniline (DAB, DADP or DATP) undergoes force-induced planarization during uniaxial elongation. Finally, Femtosecond time-resolved transient absorption (fs-TA) and electrochemical impedance spectroscopy (EIS) analysis provide direct evidence that twist effect (aromatic metrics) promotes charge transfer and effectively inhibits electron-hole recombination. **ii)** we investigated the impact of introducing different aromatic metrics, including 1,4-diaminobenzene (DAB), 4,4-diaminodiphenyl (DADP), and 4,4-diaminoterphenyl (DATP) to modulate the pore size and surface feature of the framework. Our work confirms the effective adsorption of LEV molecules on the outer surface of COFs and their penetration into the internal pore channels. Therefore, the pore size structure also plays a role in the adsorption pollutants. The adsorption capacity

order of BTT-DAB-COF < BTT-DADP-COF < BTT-DATP-COF supports this observation. Due to the steric effect, the small pore size of BTT-DAB-COF would hinder the entry of LEV molecules into the channels, resulting in lower adsorption capacity. Furthermore, the adsorption energy at the pore-site (-3.02 eV) is significantly higher than at the top-site (-0.452 eV), indicating the entrapment of LEV molecules within the pore size. The limited availability of top-site adsorption sites, due to the AA stacking mode, results in a larger quantity of adsorption sites in the pore-sites being responsible for the significant LEV adsorption by the COFs. The limited availability of top-site adsorption sites, due to the AA stacking mode, results in a larger quantity of adsorption sites in the pore sites being responsible for the significant LEV adsorption by the COFs. This above phenomenon highlights that the pore confinement effects of BTT-DATP-COF is one of the important means to regulate its catalytic activity. Subsequently, the external/internal surfacer feature of COFs was further conducted via the wetting property and the water vapor adsorption isotherms. The water contact angle of BTT-DATP-COF (136°) was much larger than that of BTT-DADP-COF (122°) and BTT-DAB-COF (60°). The water vapor adsorption isotherms confirmed that BTT-DATP-COF possesses hydrophobic pore structure. Degradation pathway results showed that the intermediate and final products were mostly hydrophilic substances (TP186, TP94 and etc.), compared with parent LEV molecule. Consequently, the hydrophilic by-products are more likely to falling off the material surface and return to the aqueous phase, contributing to the cleansing of the surface and pores of BTT-DATP-COF. The faster inner-surface renewal rate is also a significant factor contributing to the the excellent degradation performance of BTT-DATP-COFs. Moreover, hydrophobic carriers have enrichment property for organic contaminants (*Environ. Sci. Technol.*, 2022, 56, 4, 2665–2676). In summary, we believe that there is sufficient evidence to demonstrate the role of aromatic indices in COF photocatalysts.

2. For mechanism of photocatalytic decomposition of LEV: On the one hand, different scavenger quenching experiments and ESR characterization techniques confirmed that holes and some active free radicals played important roles in the system. We then demonstrated the interaction between pollutants and COF molecules and degradation path by Fukui index, DFT calculation and liquid chromatograph-mass spectrometry (LC-MS). In this photocatalysis system, h^+ , $\cdot OH$, $\cdot O_2^-$ and 1O_2 are confirmed as the primary reactive species, so Fukui index of electrophilic attack (f^-) and radical attack (f^0) are mainly considered. Specifically, N18 ($f^- = 0.2136, f^0 = 0.1129$) and N21 ($f^- = 0.0638, f^0 = 0.0318$) with high Fukui index are the most active sites for electrophilic and radical attack, indicating that the pathway II (demethylation, product $m/z=347.1389$) and pathway III (hydroxylation, product $m/z=363.1566$) should be the primary pathways for LEV degradation. The overlap between COFs and LEV bands in the PDOS reveals that the piperazine segments of LEV are more likely to combine with BTT motif, which is then oxidized by holes (h^+). It is consistent with the results of degradation pathway and the calculation of Fukui index. On the other hand, the activation mechanism of reactive oxygen species (ROS) is also considered. The results of adsorption energy confirmed that the generation of $\cdot OH$ and $\cdot O_2^-$ were located on DATP function motif (electron-acceptor unit) rather than the BTT units (electron-donor

unit). Based on the above analysis, the whole evolutive process of LEV degradation is summarized as follows: i) COFs differentially adsorb LEV molecule firstly, and then undergo the separation and migration of photoinduced electrons and holes under light irradiation; ii) the formed holes on the BTT fragments can directly oxidize the LEV; iii) in cooperation with O₂ and H₂O₂, the DATP fragment utilizes the transferred electrons for generating active radicals including •OH and •O₂⁻ to attack LEV. Spatially independent redox sites are formed within the periodic and ordered framework, which inhibits the recombination of electron-hole pairs.

1. The authors claim that BTT-DATP has a narrow FWHM compared with that of BTT-DAB-COF and BTT-DADP, which indicates that BTT-DATP shows a higher crystallinity. However, I observed the opposite result from Figure 1b-d. The XRD peak signal of BTT-DATP-COF is relatively weak. In addition, the BET surface area of BTT-DATP-COF is lower than that of other two COFs, suggesting that BTT-DATP exhibits a low crystallinity. As for peak intensity, it is related to the parameters set by the instrument.

Response: Thanks for the nice question of the reviewer. In this work, the crystallinity of three kinds of COFs is not determined by the peak intensity of PXRD. According to previous literature (*J. Am. Chem. Soc.*, 2016, 138, 4, 1234–1239; *J. Am. Chem. Soc.*, 2020, 142, 4862–4871; *Nat. Commun.*, 2023, 14, 593; *Nat. Commun.*, 2019, 10, 2467; *J. Am. Chem. Soc.*, 2017, 139, 50, 18322–1832), it was comprehensively evaluated by calculating the corresponding half-peak width (FWHM) and crystal size according to the Scherrer equation. Integrating sharp reflection, narrow half-peak width with large crystal size has the largely long-ordered domain and a very low concentration of defects. After normalizing the PXRD intensity (**Fig. S1**), the peak of BTT-DAB-COF, BTT-DADP-COF and BTT-DATP-COF at (100) plane has a FWHM of 0.47°, 0.31° and 0.25°, respectively. The microcrystal size of BTT-DAB-COF, BTT-DADP-COF and BTT-DATP-COF are 16.80, 25.88 and 32.30 nm, respectively. The BTT-DATP-COF exhibited the minimum value of FWHM and the largest domain in all of COFs. It proves that the BTT-DATP-COF has the maximum π -conjugated degree of long-ordered domains, corresponding to the higher crystallinity.

In general, the crystallinity of COF is more related to reversible bond formation and structural self-healing during the reversible reaction, and more reversible chemical reactions can improve the crystallinity (*Nature*, 2022, 604, 72-79). The value of specific surface area has no necessary relationship with the crystallinity of COFs. As typical examples of previous reports, the crystallinity of COF-5-0 is much higher than that of COF-5-50, and its specific surface area is only 1180 m²/g, lower than that of COF-5-50 (1450 m²/g) (*J. Am. Chem. Soc.* 2016, 138, 4, 1234–1239); COF_{TAPB-BTPA} shows higher crystallinity with remarkably diffraction intensity and narrower full width at half-maximum (FWHM) with respect to that of COF_{TTA-BTPA}, the former has a smaller specific surface area (1463 m²/g) than that of the latter (1842 m²/g) (*J. Phys. Chem. Lett.* 2022, 13, 6, 1398–1405). Relatively, the specific surface area of COF is more related to its micropore structure (*Environ. Sci. Technol.* 2021, 55, 5371-5381; *ACS Appl. Mater. Interfaces.* 2018, 10, 30265–30272).

Revised at Page 6 of Revised Manuscript: “After normalizing the PXRD intensity (Fig. S1), the (100) plane peak of BTT-DAB-COF, BTT-DADP-COF, and BTT-DATP-COF exhibit full width at half maximum (FWHM) values of 0.47° , 0.31° , and 0.25° , respectively. Additionally, the corresponding microcrystal sizes are measured as 16.80 nm, 25.88 nm, and 32.30 nm, respectively. According to the Scherrer equation, the sharp reflection, narrow half-peak width (FWHM) and large crystal size surface have large crystal domain and a very low concentration of defects, meaning that the BTT-DATP-COF has the maximum π -conjugated and ordered degree”.

Supplementary Figure 1. Normalized PXRD profiles of BTT-DAB-COF, BTT-DADP-COF and BTT-DATP-COF.

2. Whether BTT-DAB-COF and BTT-DADP are new COF materials? If not, please cite the related references.

Response: Thanks for the reviewer’s issue.

Added in Page 6 of Revised Manuscript, “All of BTT-DAB-COF, BTT-DADP-COF and BTT-DATP-COF have high crystallinity with the correspondingly respective (100) plane of 3.50° , 2.78° and 2.26° , respectively. The feature peaks and corresponding positions of BTT-DAB-COF and BTT-DADP-COF are consistent with the previous work (*J. Am. Chem. Soc.*, 2018, 140, 11618–11622).”

3. For the pore size (Figure 2a), how to explain the pore distribution at ~ 1.8 nm for all the COFs?

Response: Thanks for the nice question of the reviewer. The imine-linked COFs usually exhibit different kinds of pore sizes (*J. Am. Chem. Soc.*, 2022, 144, 43, 19813-19824; *J. Am. Chem. Soc.*, 2017, 139, 37, 12911-12914; *Appl. Catal. B: Environ.*, 2022, 310, 121335). In this study, Fig. S5 shows that three COFs exhibited the typical IV isotherm with an H3 hysteresis loop at $P/P_0 = 0.10-0.25$, $0.2-0.25$ and $0.25-0.35$ for BTT-DAB-COF, BTT-DADP-COF and BTT-DATP-COF, respectively, indicating its mesoporous structure. However, there is another steep slope at $P/P_0 = 0.1$, indicating some micropores structure in COF or micropores structure with pore sizes close to mesoporous. The pore distribution was mainly fitted by non-local density functional theory (NLDFT) based on N_2 adsorption-desorption data (*Nat. Commun.*, 2018, 9, 2998;

Angew. Chem. Int. Ed., 2019, 131, 4960–4964), as shown in **Fig. 2a**. The pore distribution at ~1.8 nm for all the three COFs may be due to that the three COFs are microcrystalline materials. There is existing some crystal defects as well as interstitial void among microcrystalline grains. In addition, there may be some amorphous polymers around the ordered hexagonal COFs. When N₂ adsorption-desorption experiments are conducted, this amorphous polymerization is compacted together. Integrating with the above analysis, there are two different kinds of pores appeared.

Supplementary Figure 5. N₂ adsorption-desorption isotherms of BTT-DAB-COF, BTT-DADP-COF and BTT-DATP-COF.

Fig. 2a Pore distribution of three COFs.

4. As the temperature rises, the weight of BTT-DATP-COF increased from the TGA curves. Why? Moreover, BTT-DADP-COF begins to decompose at 300 °C. So, the description of the thermal stability for these COFs is not correct.

Response: Thanks for raising the constructive suggestion from the reviewer. We have retested the thermogravimetric data of BTT-DATP-COF. The relevant description has been recorrected in our revised manuscript.

Added in Page 9 of Revised Manuscript, “Thermogravimetric analysis (TGA) results demonstrated that BTT-DAB-COF are thermally stable up to 400-500 °C under

the protection of N₂, while BTT-DADP-COF and BTT-DATP-COF begins to be decomposed at 300 °C (**Fig. S6**). The decomposition rate follows the order of BTT-DATP-COF > BTT-DADP-COF > BTT-DAB-COF.”

Supplementary Figure 6. Thermogravimetric image of BTT-DAB-COF, BTT-DADP-COF and BTT-DATP-COF.

5. In Figure S7, the lattice fringe is very clear in the local enlarge images. However, I cannot observe the lattice fringe in HRTEM images. This is very strange. Please provide the clear images.

Response: Thanks for the valuable suggestion of the reviewer. We have updated the HRTEM images, and clear lattice stripes are now provided in **Page 15 of our revised Supporting Information**.

Supplementary Figure 9. TEM image of (a) BTT-DAB-COF, (b) BTT-DADP-COF

and (c) BTT-DATP-COF (insert: the enlarged image from the white marked area)

6. In page 10, in comparison with BTT-DAB-COF, there should be an obvious blue-shift (not red shift as mentioned by the authors) of the optical absorption edge for BTT-DADP-COF and BTT-DATP-COF. Is this phenomenon consistent with the followed analysis?

Response: The reviewer's suggestions are really appreciated. In order to clarify the light absorption of COFs, the original data of UV-visible diffuse reflectance spectrum (UV-vis DRS) was further normalized for comparison. The results of UV-vis DRS were reanalyzed as follows:

By the UV-Visible diffuse reflectance spectrum (UV-Vis DRS) in **Fig. 2b**, BTT-DAB-COF exhibited relatively narrow optical absorbance with an absorption onset at 648 nm, a red-shift of the optical absorption edge was observed for BTT-DADP-COF (~10 nm) and BTT-DATP-COF (~20 nm). The corresponding energy band gaps (E_g) decreased. This suggests that terphenyl units enhance the across the plane conjugation effect (along the c plane of z axis) of the imine-linked BTT-based COFs, resulting in a narrower band gap, which is favorable for electronic transitions (*Adv. Mater.*, 2022, 34, 2203139). On the one hand, a donor-acceptor (D-A) torsional spring between BTT and aromatic aniline (DAB, DADP or DATP) undergoes force-induced planarization during uniaxial elongation leading to red-shifted absorption (*Nat. Commun.*, 2021, 12, 4243). On the other hand, reversible bond formation and structural self-healing have a central role in achieving long-range crystalline order for COFs (*Nature.*, 2022, 604, 72-79). More reversible chemistry can improve crystallinity (*Nat. Chem.*, 2013, 5, 830-834). In the presence of reversible imine linkage, the more torsion of the DATP building block than that of DAB and DADP probably strengthen the structural self-healing of COFs during the dynamic polymerization between BTT and aromatic aniline, thus achieving more long-ranged order COFs. This is consistent with the result that the order of crystallinity for BTT-DAB-COF, BTT-DADP-COF and BTT-DATP-COF.

Revised at Page 10-11 of Revised Manuscript, "By the UV-Visible diffuse reflectance spectrum (UV-Vis DRS) in **Fig. 2b**, BTT-DAB-COF exhibited relatively narrow optical absorbance with an absorption onset at 648 nm, a red-shift of the optical absorption edge was observed for BTT-DADP-COF (~10 nm) and BTT-DATP-COF (~20 nm). The corresponding energy band gaps (E_g) decreased from 2.21, to 2.18, and 2.15 eV (**Fig. S10**). This suggests that terphenyl units enhance the conjugation effect across the plane (along the c plane of z axis) of the imine-linked BTT-based COFs, resulting in a narrower band gap, which is favorable for electronic transitions³². A donor-acceptor (D-A) torsional spring between BTT and aromatic aniline (DAB, DADP or DATP) undergoes force-induced planarization during uniaxial elongation leading to red-shifted absorption³³. This is consistent with the result that BTT-DATP-COF has the highest crystallinity, as well as the result of (001) plane shift."

Fig. 2a UV/vis DRS spectra and the Kubelka–Munk-transformed reflectance spectra.

7. In page 12, I observe that BTT-DATP-COF shows a slow adsorption kinetic among the COFs studied here, as seen in Figure 2c. The adsorption kinetic (k_2) of BTT-DATP-COF was 0.000426 g.mg/s, less than 60-fold that of DTT-DAB-COF (0.02551 g.mg/s). In addition, the calculated mass-transfer coefficient (k_f) values for the LEV are also inconsistent with the absorption kinetics results. So, this result is not enough to support the conclusion of the manuscript.

Response: Thanks for the reviewer's issue. Details are elaborated as follows:

In this work, the adsorption capacity of LEV by the three COFs determined by adsorption kinetics and adsorption isotherms followed the order of BTT-DAB-COF < BTT-DADP-COF < BTT-DATP-COF. Specifically, the saturated adsorption capacity (q_m) of BTT-DATP COF calculated according to Langmuir isotherm model was 86.65 mg/g, which was much higher than that of BTT-DADP-COF (32.30 mg/g) and BTT-DAB-COF (29.99 mg/g). Obviously, the adsorption capacity is positively correlated with the pore size, independent of the BET-surface area values of COFs. Although BTT-DATP-COF has larger adsorption capacity, the smallest adsorption kinetic coefficient is obtained. Based on the specific phenomenon, the wetting property of three COFs was further conducted. The water contact angle of BTT-DATP-COF (136°) was much larger than that of BTT-DADP-COF (122°) and BTT-DAB-COF (60°), indicating the BTT-DATP-COF had difficult dispersion into water in a short time due to the hydrophobic effect. Due to the steric effect, the small pore size of BTT-DAB-COF would prevent LEV molecules from entering the channels to obtain a low accumulated capacity, and quickly reaching the adsorption equilibrium onto the external surface. Since all coefficients related (k_2 and k_f) to the time of adsorption equilibrium, BBT-DAB-COF has the highest coefficient. Moreover, hydrophobic carriers have enrichment property for organic contaminants (*Environ. Sci. Technol.*, 2022, 56, 4, 2665–2676). The adsorption capacity order of BTT-DAB-COF < BTT-DADP-COF < BTT-DATP-COF may also be due to the π - π interaction with the benzene ring. Therefore, it can be found that the adsorption capacity of the three COFs for organic pollutants is primarily related to the integrity of pore size and π - π interaction.

Subsequently, we discussed the relationship between hydrophobicity and degradation products. The water vapor adsorption isotherms confirmed that BBT-DATP-COF has a hydrophobic pore structure. Degradation pathway results showed

that the intermediate and final products were mostly hydrophilic substances (TP186, TP94 and etc.), compared with parent LEV molecule. Thus, the hydrophilic by-products are more likely to falling off the material surface and return to the aqueous phase, contributing to clean the surface and pores of BTT-DATP-COF. The faster inner-surface renewal rate is also one of the important reasons for the excellent degradation performance of BTT-DATP-COFs.

In a word, these results are enough to support the conclusion that BTT-DATP-COF has more excellent degradation in this study.

8. Why use aqueous HClO₄ to adjust the initial pH value? As we know, HClO₄ is a strong oxidant, which can oxidize and degrade pollutants.

Response: Thanks for the valuable suggestion of the reviewer. In order to eliminate this effect, we re-adjust the pH value of the system with hydrochloric acid (HCl) and sodium hydroxide (NaOH) (Fig. 4a). **The image was revised at Page 22 of Revised Manuscript.**

Fig. 4a Photocatalytic performance for LEV degradation with different pH.

Previously, we applied HClO₄. It is because HClO₄ is an inert acid which is generally used for solution pH adjusting in organic pollutant oxidation systems (*Environ. Sci. Technol.*, 2021, 55, 10, 7034-7043; *Environ. Sci. Technol.*, 56, 7, 4367-4376; *Water Res.*, 2021, 207, 117796), such as photocatalysis and advanced oxidation processes (AOPs). If HNO₃ and H₂SO₄ are applied, possible oxidation of pollutants may occur under radical-induced reactions.

9. Please provide the detection method for the concentrations of F⁻, NO₃-N and DON after reaction.

Response: Thanks for the reviewer's issue. The F⁻ was determined by ion chromatography (HJ 84-2016), the NO₃-N was extracted by potassium chloride solution and double wavelength colorimetric method (HJ 634-2012), and the DON was extracted by potassium chloride solution, an indirect method shown in HJ 636-2012.

Added in Page 7-8 of Revised Supporting Information

The measured procedure of nitrate nitrogen (NO₃-N): First, NO₂-N in potassium chlorate solution was extracted from the solution. Under acidic conditions,

NO₂-N in the extraction solution would react with sulfanilamide to form diazo salt, and then combined with N- (1 naphthyl) -ethylenediamine hydrochloride to form red dye, which had a maximum absorption at 543 nm. NO₂-N concentration and absorbance conformed to Lambert-Beer law in a certain range. Secondly, NO₂-N and NO₃-N in the solution were extracted from the potassium chlorate solution, and the extracted solution reduced NO₃-N to NO₂-N through the reduction column. The total amount of NO₃-N and NO₂-N was detected according to the above method. The concentration of NO₃-N was the difference between the total amount of NO₃-N and NO₂-N and the content of NO₂-N.

The measured procedure of dissolved organic nitrogen (DON): The DON was determined by indirect method, that is, DON= total nitrogen (TN)- nitrate nitrogen (NO₃-N) - ammonium nitrogen (AN). For TN (HJ 636-2012) detection: Appropriate number of samples was taken and solution pH was adjusted to 5–9 with NaOH or H₂SO₄ solution. Then, 10 mL of the above sample was taken into 25 mL stop-ground glass colorimetric tube, and 1 mL HCl solution was added. Afterwards, the mixture was diluted to 25 mL with water, and a 10 mm quartz colorimetric dish was used on an ultraviolet spectrophotometer with water as reference to determine the absorbance at 220 and 275 nm, respectively. For AN (HJ 535-2009) detection: After dechlorination and pre-distillation, the absorbance of the sample was measured in a 20 mm cupola at a wavelength of 420 nm, with water as the reference.

The measured procedure of fluoride ion (F⁻): After instrument calibration with a standard liquid of fluoride ions, the appropriate amount of blank sample (pure water) and the sample were measured by an ion chromatograph (IC) for detection. Separation column: IonPaC AS23 anion separation column (1.8 mM sodium carbonate) + IonPac AG23 anion protection column (1.7 mM sodium bicarbonate), and the flow rate was 1.0 mL/min.

10. The authors claim that TP186 and TP94 are the main intermediates. It is recommended to use the COF as photocatalyst to degrade TP186 and TP94, respectively. If the final products can be obtained by using this method?

Response: Thanks for the reviewer's issue. TP186, a precursor to TP94, has not the standard substance with the corresponding CAS number at present. Therefore, the degradation performance of BTT-DATP-COF for TP94 was only studied here.

TP94, standing for phenol, is the late-stage products of LEV degradation from the degradation pathway. As shown in **Fig. R3**, the removal efficiency of phenol is 34 % for BTT-DATP-COF within two hours. The slow degradation of phenol by BTT-DATP-COF belongs to the characteristics of late-stage degradation products. In other words, when the material degraded LEV to phenol, the degradation rate slowed down and the residual part in the solution was then detected.

Fig. R3 Photocatalytic activity for phenol degradation by BTT-DATP-COF under visible-light illumination.

11. In Figure S17, what is the concentration of inorganic anions and cations?

Response: Thanks for the reviewer's issue. To determine the effect of coexisting anion (Cl^- , NO_2^- , NO_3^- and SO_4^{2-}) and cations (Na^+ , K^+ , Cu^{2+} and Al^{3+}) species, a concentration of 10 mM for each of anions and cation was introduced into antibiotics levofloxacin hydrochloride solution (**Fig. S30**). **We claim this in the revised Supporting Information at Page 30.**

Supplementary Figure 30. The visible light driven photocatalytic performance of BTT-DATP-COF for the degradation of LEV with influence of (a) anion and (b) cation species. (Reaction condition: LEV =10 mg/L; catalyst dose =0.1 g/L; ionic concentration =10 mM; T =25°C; 300 W Xe lamp with $\lambda > 420$ nm)

12. The authors studied the influence of HA on the adsorption capacity. Why select HA? Can other aromatic acid promote this process?

Response: Thanks for the reviewer's issue. The presence of coexisting constituents, such as dissolved natural organic matter (NOM), can dramatically decrease the degradation efficiency of target pollutants. NOM has been widely recognized as an inhibitor in photocatalysis because of its ability to scavenge photogenerated holes and radicals, interfere with charge transfer, and occlude the sites of ROS generation (*Environ. Sci. Technol.*, 2012, 46, 11, 6228–6235). From the perspective of practical wastewater treatment, it is essential to evaluate the impact of NOM on photocatalytic degradation of organic pollutants. Humic acid (HA) is a major component of NOM. It

is the most frequently existing organic compounds in water body and possesses abundant phenolic, hydroxyl, or carboxylic groups. It can form a multi-media system when it coexists with pollutants, which can complicate the migration and transformation behavior of pollutants in the environment through non-specific processes including electrostatic interaction, hydrogen bonding, dipole interaction and hydrophobic effect. Thus, HA is the most widely used model NOM to study the influence on organic pollutants degradation in photocatalytic system.

In line with the suggestions of reviewers, other two typical aromatic acids (*i.e.*, fulvic acid and sodium humate) were additionally introduced into the system to investigate the impact on LEV degradation efficiency.

Added in Page 21 of Revised Manuscript, “The addition of natural humic acid (HA), sodium humic acid (HA-Na) and fulvic acid (FD) greatly improved the adsorption capacity (**Fig. S31a**). Meanwhile, all the three NOMs slightly reduced the removal efficiency of LEV by BTT-DATP-COF within 120 min, and negligible effect was observed even with the increase of HA concentration (**Fig. S31b**). It contributes to the bridging function of NOM between LEV and BTT-DATP-COF (*App. Catal. B: Environ.*, 2020, 262, 118308), detailed analysis is provided in the Supporting Information.”

Added in Page 31 of Revised Supporting Information, “NOM can act as a bridge between LEV and BTT-DATP-COF due to the variety of functional groups, and the side of hydrophobic part can combine with COFs via π - π interaction to form COF-NOM aggregation. Then, the hydrophilic part of aggregation can immobilize LEV via hydrophilic interaction to form COF-NOM-LEV, thus promoting the adsorption of LEV (*Appl. Catal. B: Environ.*, 2022, 262, 118308). With the increasing concentration of HA, the degradation efficiency of LEV was slightly restrained, assigning to the competitive interaction of HA with reactive species and light-screening effects induced by the conjugated double bonds chromophore structure of HA in the visible region (*J. Hazard. Materials.* 2021, 417, 126034; *Water. Res.*, 2022, 219, 118558).

Supplementary Figure 31. The visible light driven photocatalytic performance of BTT-DATP-COF for the degradation of LEV with influence of dissolved natural organic matter (a) and humic acid (b)

13. It is suggested to unify the name of material. The name of COFs in Figures S4, S5, and S21 are not consistent with that in other Figures.

Response: Thanks for the reviewer's issue. We re-checked the overall manuscript and accordingly ensure the names of all the COFs are consistent in our revised manuscript.

14. In Figure S29, the adsorption energy between BTT and O₂ (0.583 eV) is higher than that of DATP and O₂ (0.544 eV). Why do the authors think that the O₂ molecule can be more easily adsorbed to the DATP function motif rather than the BTT units?

Response: Thanks for the reviewer's issue. The O₂ adsorption and activation onto photocatalyst are investigated through density functional theory (DFT) calculation. The larger the adsorption energy is, the more energy the material needs to obtain for effective O₂ adsorption. Conversely, if the adsorption energy is smaller, the antibonding orbitals of O₂ can be populated by the photogenerated electrons along with the O-O bond being easily stretched to form active species such as superoxide radicals (*J. Am. Chem. Soc.*, 2013, 135, 15750–15753; *Energy Environ. Sci.*, 2022, 15, 830-842). Thus, there is a stronger adsorption capacity and surface binding strength between material and O₂.

In this work, the O₂ molecule can be more easily adsorbed to the DATP function motif (electron-acceptor unit) rather than the BTT units (electron-donor unit). From **Fig. S44**, the adsorption energy and bond length after activation is 0.544 eV and 1.247 Å on DATP units, lower than that of the BBT units (0.583 eV and 1.245 Å). In this way, the excited electron from BTT unit transferred to the adsorption site (in DATP units) could lead to the generation of $\cdot\text{O}_2^-$.

Supplementary Figure 44. The favorite adsorption configurations of O₂ and H₂O₂

molecule adsorbed on BTT-DATP-COF.

15. As reported previously (J. Am. Chem. Soc. 2013, 135, 546–549, J. Am. Chem. Soc. 2018, 47, 16124–16133, and Angew. Chem. Int. Ed. 2013, 52, 13052–13056), the interlayer interaction affects the stacking behavior between layers, thus significantly controlling the crystallinity, porosity, and stability of the COFs. Why BTT-DATP-COF with largest torsional angle has smallest distance of interlayer stacking and best crystallinity.

Response: The nice question from the reviewer is really appreciated. We have carefully read the three literatures proposed by the reviewer and cited to support the findings of our work. It was found that the interlayer stacking behavior was affected by the introducing substituents, which thus significantly controlled the crystallinity, porosity, and stability of the COFs. The details are as follows: i) The introduction of alkoxy (-OCH₃) in COFs leads to the loss of planarity and structural rigidity between the phenyl rings during the reversible synthesis process of COFs, decreasing the stacking between 2D layers and subsequently reducing the crystallinity (Angew. Chem. Int. Ed. 2013, 52, 13052–13056); ii) The alternating arrangement between fluoro-substituted groups and non-substituted aromatic units is conducive to the π -electron interlayer complementarity, increasing interlayer stacking and improving crystallinity (J. Am. Chem. Soc., 2013, 135, 546–549). iii) In AA stacking mode, bulk alkyl substitution (ethyl and isopropyl) is more likely to produce steric hindrance, resulting in the close repulsion between adjacent layers and lower crystallinity (J. Am. Chem. Soc., 2018, 47, 16124–16133).

In this work, three building blocks (DAB, DADP and DATP) have no substituents, and these torsion angles were calculated relative to the amino component. Although the triphenyl has the greatest torsion angle, it is a flexible monomer that has been shown to be more efficient at stacking during polymerization reactions (J. Am. Chem. Soc. 2022, 144, 34, 15581–15594). Therefore, we deduce that there are two reasons for the smallest distance of interlayer stacking and best crystallinity of BTT-DATP-COF. Both reversible bond formation and structural self-healing have central role in achieving long-range crystalline order for COFs (Nature., 2022, 604, 72–79). More reversible chemistry can improve crystallinity (Nat. Chem., 2013, 5, 830–834). In the presence of reversible imine linkage, the more torsion of the DATP building block than that of DAB and DADP probably strengthens the structural self-healing of COFs during the dynamic polymerization between BTT and aromatic aniline, thus achieving more long-ranged order COFs. This is consistent with the result that the order of crystallinity for BTT-DAB-COF < BTT-DADP-COF < BTT-DATP-COF.

We further discussed this mechanism in the Revised Manuscript at Page 7: “Furthermore, DAB, DADP and DATP building block have torsion angles relative to the amino component, corresponding to 28.5°, 145.8° and 146.9°, respectively. Although the triphenyl has the greatest torsion angle, it is a flexible monomer that has been shown to be more efficient at stacking during polymer reactions. Reversible bond formation and structural self-healing have a central role in achieving long-range crystalline order for COFs. In the presence of reversible imine linkage, the more torsion

of the DATP building block than that of DAB and DADP probably strengthen the structural self-healing of COFs during the dynamic polymerization between BTT and aromatic aniline, thus achieving more long-ranged order COFs. This is consistent with the result that the crystallinity order of BTT-DAB-COF < BTT-DADP-COF < BTT-DATP-COF. It verifies that the difference of torsion angle in linkers varies the crystallinity, conjugation, and long-range order of COFs.”

16. Authors should use the MOTT to verify the conduction bands, and use the UPS analyses to determine the band position in addition to the XPS analyses.

Response: Thanks for the reviewer’s issue. Based on the reviewer's comments, the MOTT and UPS data for the material were provided in our revised manuscript.

Considering the high energy of the light source used by XPS-VB, the main research is the deep level of electronic information. The UPS is mainly concerned with the low binding energy electronic information, that is, the electronic information of the shell layer. When detecting the valence band electronic information of materials, the use of ultraviolet lamp sources is sufficient (*Angew. Chem. Int. Ed.*, 2022, e202218868; *J. Am. Chem. Soc.*, 2016, 138, 8928–8935; *Angew. Chem. Int. Ed.*, 2017, 129, 4270–4274; *Nat. Energy.*, 2019, 4, 690–699).

Added in Page 11 of Revised Manuscript, “On the basis of ultraviolet photoelectron spectroscopy (UPS) (**Fig. S11**), the relative valence band maximum (VBM) of BTT-DAB-COF, BTT-DADP-COF and BTT-DATP-COF is calculated to be 2.19, 2.33 and 2.37 eV (vs. Relative Hydrogen Electrode-RHE) (*Angew. Chem. Int. Ed.*, 2022, e202218868; *Nat. Commun.*, 2022, 13, 1355). We further carried out the Mott-Schottky electrochemical measurements to determine the energy band position (**Fig. S12**), the conduction band (CB) of BTT-DAB-COF, BTT-DADP-COF and BTT-DATP-COF are calculated to be -0.10 eV, -0.03 eV, and 0.02 eV (vs. RHE) and -0.45, 0.4 and -0.35 eV (vs. Normal Hydrogen Electrode-NHE), respectively. Meanwhile, the VB positions of BTT-DAB-COF, BTT-DADP-COF and BTT-DATP-COF are 2.11 eV, 2.15 eV and 2.17 eV (vs. RHE), respectively, which is not much different from the values obtained by UPS.”

Supplementary Figure 11. UPS spectrum of the three COFs.

Supplementary Figure 12. Mott-Schottky plot of the three COFs (0.1M NaSO₄ solution, pH=6.2).

17. Authors claimed that hydrophobic effect can promote the transport of degraded substances out of the pore and has a better self-purification effect, however, the result of water contact angle can only prove the hydrophobicity in external surface, which can not verify hydrophobic property of the pores.

Response: Thanks for raising a nice question. To study the internal surface characteristics of pore in COFs, water vapor adsorption-desorption isotherms were measured at 298 K (*J. Am. Chem. Soc.*, 2013, 135, 14, 5328–533; *Angew. Chem. Int.*

Ed., 2021, 60, 21838–2184; *J. Am. Chem. Soc.*, 2015, 137, 7217–7223 and *Nat. Commun.*, 2021, 12, 6747).

Added in Page 29 of Revised Supporting Information, “As shown in **Fig. S28**, BTT-DAB-COF, BTT-DADP-COF and BTT-DATP-COF can adsorb water with 5.018, 2.608 and 2.909 mmol/g uptake at $P/P_0 = 1$, respectively. This phenomenon is consistent with previous literature. That is, HFPTP-DMePDA-COF has a hydrophobic inner channel due to the water vapor adsorption capacity of greatly decreasing, despite having a larger pore size (*Nat. Commun.* 2021, 12, 6747). The very different adsorption-desorption isotherms for water confirm the hydrophobic nature of pore channels of BTT-DATP-COF compared with BTT-DAB-COF. Besides, these COFs display a larger hysteresis loop to complete the adsorption-desorption exchange cycle, which is a feature of mesoporous COFs. Therefore, it is deduced that the hydrophilic by-products are more likely to falling off the material and return to the aqueous phase, contributing to clean the surface and pores of BTT-DATP-COF.”

Added in Page 19 of Revised Manuscript, “The water vapor adsorption isotherms confirmed that BTT-DATP-COF has a hydrophobic pore structure (**Fig. S28**), that is, with a hydrophobic inner surface.”

Supplementary Figure 28. Water vapor adsorption-desorption isotherms of three COFs measured at 298 K.

18. The authors claimed that “increase of crystal size with more ordered and conjugated structure is conducive to charge carrier transport via inhibiting carrier recombination”. Why the very different crystal size of the COFs (16.80, 25.88 nm) exhibit similar carrier lifetime (0.3, 0.31 ps)?

Response: Thanks for the reviewer’s issue. We retested and reanalyzed the fs-TA of BTT-DAB-COF and BTT-DADP-COF, and uploaded the raw TA data accompanying by the revised manuscript for reference. The description was corrected as follows:

Added in Page 24 of Revised Manuscript, “The fitting dynamics reveal the variation of two decay time constants in **Fig. R4**, and found that the τ_1 and τ_2 of BTT-DAB-COF (0.6 ps) have no change significantly, due to no excited state absorption and only having one component attenuation. The average lifetime of BTT-DATP-COF (75 ps) and BTT-DADP-COF (64 ps) is much longer-lived than BTT-DAB-COF (3.23 ps) due to the generally longer lifetime of excimer formation and decay compared to singlet exciton emission. The results of carrier lifetime are consistent with the crystal sizes of

BTT-DAB-COF, BTT-DADP-COF and BTT-DATP-COF, corresponding to respective 16.80, 25.88 and 32.30 nm (Note S1). That is, the increase of crystal size, coupled with a more ordered and conjugated structure, effectively suppresses carrier recombination (*Adv. Mater.*, 2020 32, 2003965).

Fig. R4 (a-c) 2D mapping TA spectra of BTT-DAB-COF, BTT-DADP-COF and BTT-DATP-COF, (d-f) TA spectra signals of three COFs on the fs-ns timescales and (g-i) TA kinetics traces probed of three COFs.

19. Authors are strongly suggested to give more evidences to support that the twist effect of terphenyl can inhibit electron-hole recombination.

Response: Thanks for the valuable suggestion of the reviewer. Various characterizations including steady-state photoluminescence spectroscopy (PL), femtosecond time-resolved transient absorption spectroscopy (fs-TA), and electrochemical impedance spectroscopy (EIS) of three COFs were used to demonstrate the twist effect of terphenyl can inhibit electron-hole recombination.

In the original manuscript, we demonstrated that BTT-DATP-COF had significant fluorescence quenching by steady-state photoluminescence spectroscopy (PL), which indicated that BTT-DATP-COF has minimal photogenerated electron-hole recombination under light conditions (Fig. S35). Subsequently, femtosecond time-resolved transient absorption spectroscopy (fs-TA) gave the most intuitive evidence via studying photoexcited carrier dynamics under the influence of twist effect of terphenyl in these COFs. As shown Fig. 4R a-c, TA spectra with different time delays under pulse

excitation were recorded. BTT-DAB-COF showed obvious ground state bleach (GSB) signals at 525 nm and stimulated emission (SE) around 650 nm, and basically recovered within 10 ps (**Fig. 4R d**). The GSB signal was caused by neutral singlet excitons, associated with characteristic stimulated emission. In contrast, BTT-DADP-COF and BTT-DATP-COF exhibited extensive excited state absorption (ESA) feature that developed on ultrafast time scales (**Fig. 4R e-f**). The ESA with the center of 500 nm and 650 nm appeared immediately after the photoexcitation of BTT-DATP-COF, attributing to excimer formation (electron excitation). Interesting, a negative peak can be observed at 750 nm for BTT-DADP-COF and BTT-DATP-COF, which are reasonably interpreted as a photoinduced emission process originating from the interfacial charge transfer (labelled as ICT signal) from DADP/DATP to BTT units (*Angew. Chem. Int. Ed.*, 2003, 62, e202218688). The above phenomena provide direct evidence that torsion effect promotes charge transfer and effectively inhibits electron-hole recombination. Previous studies have shown that the optical density (ΔOD) signal amplitude is proportional to the exciton population. It infers that the number of excitons for BTT-DATP-COF and BTT-DADP-COF is much more than that of BTT-DAB-COF. The fitting dynamics reveal the variation of two decay time constants in **Fig. 4R g-i**, and found that the τ_1 and τ_2 of BTT-DAB COF (0.6 ps) have no change significantly, due to no excited state absorption and only having one component attenuation. The average lifetime of BTT-DATP-COF (75 ps) and BTT-DADP-COF (64 ps) is much longer-lived than BTT-DADP-COF (3.23 ps) due to the generally longer lifetime of excimer formation and decay compared to singlet exciton emission. The results of carrier lifetime are consistent with the crystal sizes of BTT-DAB-COF, BTT-DADP-COF and BTT-DATP-COF, corresponding to respective 16.80, 25.88 and 32.30 nm (**Note S1**). That is, the increase of crystal size, coupled with a more ordered and conjugated structure, effectively suppresses carrier recombination (*Adv. Mater.*, 2020, 32, 2003965). Considering DATP with a certain flexibility and twist effect (*J. Am. Chem. Soc.*, 2022, 144, 15581-15594), it can be inferred that upon absorbing a certain amount of energy from visible light irradiation, BTT-DATP-COF contribute to a free rotation between DATP and imine bond together at the arm of framework, which then accelerate the migration of carriers through the kinetic energy conversion and inhibiting electron-hole recombination. Based on this, we confirmed that the twist effect not only controls the crystallinity and conjugation during the synthesis of COFs, but also is a prerequisite for establishing longitudinal and transverse charge-transfer channels between terphenyl and BTT motif.

Supplementary Figure 35. PL spectra of BTT-DAB-COF, BTT-DADP-COF and BTT-DATP-COF.

Fig. R4 (a-c) 2D mapping TA spectra of BTT-DAB-COF, BTT-DADP-COF and BTT-DATP-COF, (d-f) TA spectra signals of three COFs on the fs-ns timescales and (g-i) TA kinetics traces probed of three COFs.

Finally, the interfacial properties between the electrolyte and the electrode were characterized via electrochemical impedance spectroscopy (EIS). The EIS spectra of the three COFs were retested, and the fitting results of the data are shown in Supplementary Figure 32. It can be found that the high-frequency impedance of BTT-DADP-COF and BTT-DATP-COF were significantly lower than that of BTT-DAB-COF. The impedance spectrum of the high-frequency part is given as an illustration in

the upper right corner of the figure. Compared with BTT-DADP-COF, the BTT-DATP-COF had a similar high-frequency impedance, but the low-frequency part was significantly smaller, which was conducive to the transmission of photogenerated carriers on the material surface. The three-phase analog circuit used for fitting is shown in **Fig. S37**, and the fitting results are given in **Table S9**. Specifically, it can be seen that BTT-DATP-COF exhibited the lowest charge-transfer resistance (R_{CT} , $3.52 \times 10^5 \Omega$), and meanwhile, it also exhibited the low internal resistance (R_S , 11.30Ω) and interfacial resistance (R_{SC} , 45.88Ω). Thus, it is indicated the higher electron mobility in the former material, and the electron-hole recombination rate being limited more effectively by the twist effect.

Supplementary Table 9. Summary of the EIS fitting results of impedance data for the three COFs.

Materials	Parameters ^a				
	R_S (Ω)	R_{SC} (Ω)	R_{CT} (Ω)	C_1 (F)	C_2 (F)
BTT-DAB-COF	408.20	6093.00	1.23×10^6	4.12×10^{-8}	2.02×10^{-7}
BTT-DADP-COF	11.19	44.59	4.88×10^5	5.50×10^{-9}	2.25×10^{-5}
BTT-DATP-COF	11.30	45.88	3.52×10^5	5.50×10^{-9}	2.20×10^{-5}

^a R_S is internal resistance; R_{SC} is interfacial resistance; R_{CT} is charge-transfer resistance; C_1 and C_2 are Constant Phase Element (CPE), which are ideal capacitance in analog circuits.

Supplementary Figure 37. EIS spectra of the three COFs. (the illustration of the enlarged diagram of material internal resistance and the simulated circuit diagram).

Reviewer #3 (Remarks to the Author):

This manuscript described the synthesis of three COF materials, which were further used as photocatalysts. Characterizations have been made to analyze the COFs and to evaluate their performance for pollutant photodegradation. Extensive computational studies have been performed to understand the materials as photocatalysts. Although the material design does not stand out from many other works in this field, the fundamental knowledge obtained by the detailed analysis should be of slight interest for the readers of Nature Communications. In addition to some flaws and claims without clear support, the key concerns are about the torsion angles and stacking behavior in the COF models, and the relationship between hydrophobicity of the materials and their performance. Therefore, I recommend reconsidering this manuscript after a revision, provided that the major findings and conclusions still hold true. The detailed comments and questions are listed as following.

Response: The reviewer's valuable time and effort on our manuscript are greatly appreciated. We have made major revisions according to the reviewer's suggestion.

1. The authors highlighted the different torsion angles of the aromatic anilines in the title, introduction (line 97) and Figure 1. However, it is unclear how the torsion angles are obtained. As this manuscript includes extensive computational studies, which use the structural models of the COFs, torsion angles are important and they could affect the computational results.

Response: Thanks for the reviewer's issue. According to the previous reports (*J. Am. Chem. Soc.* 2020, 142, 21, 9752–9762) shown in **Fig. R5**, the aromatic anilines fragments were chosen as the model. To obtain the optimized geometry, we performed potential energy scanning with multiple rounds of calculations for different torsion angles using the DFT method. We have addressed the different torsion angles of the aromatic anilines of different building blocks in our study by measuring the dihedral angles using adjacent four atoms.

In addition, the periodic DFT calculations are performed for the following adsorption behavior of different species onto COFs, which also keeps consistent with the previous reference (*Angew. Chem. Int. Ed.*, 2022, 61, e202202328).

FIGURE REDACTED

Fig. R5 The referenced literature for the calculation of torsional angles in aromatic anilines.

2. The authors used simulated PXRD patterns to justify the stacking behaviors of the COFs. However, with very few peaks in the experimental PXRD pattern, it is difficult to distinguish different models, e.g., BTT-DADP-COF. In addition, AA inclining and partial AB stacking are other common stacking behavior of 2D COFs. The authors need to add more characterizations such as in-plane TEM images to justify the models.

Response: Thanks for the valuable suggestion of the reviewer. Distinct in-plane TEM images are provided in **Fig. S8**.

Supplementary Figure 8. High-resolution TEM image of (a) BTT-DAB-COF, (d) BTT-DADP-COF and (g) BTT-DATP-COF; Enlarged reversed fast FFT image from

the yellow marked area (b) BTT-DAB-COF, (e) BTT-DADP-COF and (h) BTT-DATP-COF; the diffraction pattern of real space FFT (c) BTT-DAB-COF, (f) BTT-DADP-COF and (i) BTT-DATP-COF.

Added in Page 10 of Revised Manuscript, “As shown in **Fig. S8**, the high-resolution transmission electron microscopy (HRTEM) image of BTT-DAB-COF, BTT-DADP-COF and BTT-DATP-COF revealed the formation of a highly ordered porous network with distinct and ordered honeycomb-like pore. Furthermore, the clearly periodic hexagonal pore can be observed by high-angle annular dark-field transmission electron microscopy (HAADF-TEM), and the angle of relative orientation between neighboring pores closely approximates 60° . It possibly proved that AA stacking occurred in COFs (*J. Am. Chem. Soc.* 2023, 145, 15, 8364–8374).”

3. As the authors mentioned, the Scherrer equation can give indication of the size of crystalline domains. Therefore, the authors should also discuss the crystal size of the COFs before draw conclusion of the ordered degree of their π -conjugation system.

Response: The reviewer’s suggestions are really appreciated, and we have discussed the relationship between the crystalline size and the ordered degree of their π -conjugate system in our revised manuscript.

Added in Page 6 of Revised Manuscript, “After normalizing the PXRD intensity (**Fig. S1**), the (100) plane peaks of BTT-DAB-COF, BTT-DADP-COF, and BTT-DATP-COF exhibit full width at half maximum (FWHM) values of 0.47° , 0.31° , and 0.25° , respectively. Additionally, the corresponding microcrystal size are calculated as 16.80 nm, 25.88 nm, and 32.30 nm, respectively. According to the *Scherrer* equation, the sharp reflection, narrow half-peak width (FWHM) and large crystal size indicate large ordered domain and very low concentration of defects, meaning that the BTT-DATP-COF has the maximum π -conjugated and ordered degree.”

Supplementary Figure 1. Normalized PXRD profiles of BTT-DAB-COF, BTT-DADP-COF and BTT-DATP-COF.

4. It is difficult to see the 001 reflection in Figure 1. I suggest to add a zoom-in figure in the SI to show that region.

Response: Thanks for the valuable suggestion of the reviewer. A zoom-in figure of the

(001) reflection region has been added to the revised manuscript.

Added in Page 6 of Revised Manuscript, “The wide peak at 25.21°, 25.96° and 26.44° can be assigned to the (001) plane generated by π stacking of 2D layers for the three COFs (Fig. S2).”

Fig. S2 was added in Page 11 of Revised Supporting Information.

Supplementary Figure 2. XRD pattern of three COFs (the inset shows the location of the broad peak of COFs)

5. In line 122, the authors described “All of BTT-DAB-COF, BTT-DADP COF and BTT-DATP-COF have high crystallinity with the correspondingly respective (100) plane of 2.26°, 2.78° and 3.50°, respectively.” The text is inconsistent with the overall description of the COFs, where BTT-DAB-COF should have the smallest unit cell, and thus the largest 2theta angle of the 100 reflection.

Response: Thanks for the reviewer’s issue. We carefully re-checked the original data and uploaded the PXRD data accompanying by the revised manuscript for reference. Specifically, the diffraction angles of (100) planes of BTT-DAB-COF, BTT-DADP-COF and BTT-DATP-COF are 3.50°, 2.78° and 2.26°, respectively (Fig. S2). We have corrected this expression in the revised manuscript.

Supplementary Figure 2. XRD pattern of three COFs (the zoom-in figure shows the location of the broad peak of COF)

6. In line 247, the authors mentioned “..., appropriate pore size and hydrophobic effect can promote the transport of degraded substances out of the pore, and has a better self-purification effect”. It is clear that large pore size can facilitate diffusion, but the authors did not discuss what “self-purification effect” means, and how hydrophobicity can promote such effect, and how the effect is related to the performance of LEV

degradation.

Response: Thanks for raising a nice question from the reviewer. In our submitted manuscript, we discuss that pore size and hydrophobicity together promote the self-purification effect, thereby improving the degradation performance of LEV.

The “self-purification effect” means that due to the hydrophilic/hydrophobic relationship, the degradation by-products on the internal/external surface of the material are more likely to fall off from the active site and return to the water phase. With the help of favorable transport channel of BTT-DATP-COF, by-products could be transferred out in time to achieve the purpose of cleaning the surface and pores of the material.

By analyzing the transformation products (TPs) detected by liquid chromatography-mass spectrometry (LC-MS), we found that the intermediate and final products were mostly hydrophilic substances (TP186, TP94 and etc.), compared with parent LEV molecule. In contrast, the water vapor adsorption isotherms confirmed that BBT-DATP-COF has a hydrophobic pore structure (**Fig. S28**), that is, with a hydrophobic inner surface. Therefore, the hydrophilic by-products are more likely to falling off the hydrophobic surface of BTT-DATP-COF and return to the aqueous phase, releasing the active sites on the surface of material. Then, the degradation products are transferred out of the pore in time with the help of favorable transport channel of BTT-DATP-COF to achieve the purpose of cleaning the surface and pores. We could infer that the faster inner-surface renewal rate is also one of the important reasons for the excellent degradation performance of BTT-DATP-COFs.

Added in Page 19 of Revised Manuscript, “Degradation pathway results showed that the intermediate and final products were mostly hydrophilic substances (TP186, TP94 and etc.) compared with parent LEV molecule. The water vapor adsorption isotherms confirmed that BBT-DATP-COF has a hydrophobic pore structure (**Fig. S28**), that is, with a hydrophobic inner surface. Therefore, the hydrophilic by-products are more likely to falling off the hydrophobic surface of BTT-DATP-COF and return to the aqueous phase, releasing the active sites on the surface of material. With the help of favorable transport channel of BTT-DATP-COF, by-products could be transferred out in time to further achieve the purpose of cleaning the surface and pores of the material. We could infer that this “self-purification effect” promotes a faster inner-surface renewal rate, which is one of the important reasons for the excellent degradation performance of BTT-DATP-COFs.”

7. To highlight the role of the COFs for the photodegradation of LEV, the authors need to show an experiment without using the COF materials as a reference.

Response: Thank you very much for pointing out this problem.

Added in Page 14 of Revised Manuscript, “Furthermore, almost no photolysis of LEV occurred under visible-light irradiation in the absence of COFs materials (**Fig. S19**).”

Supplementary Figure 19. Photocatalysis of LEV without COFs under visible-light illumination.

8. As photocatalysts, how much quantum efficiency do the COFs provide?

Response: The reviewer’s suggestions are really appreciated, and the quantum efficiency of BTT-DATP-COF have provided in our revised manuscript.

Added in Page 5 of Revised Supporting Information for quantum efficiency analysis method, “In this photocatalysis system, the apparent quantum efficiency (AQE) is defined as the ratio of electron transfer number to photon injection number during photocatalysis (Equation 5-9). In the visible light regions, six representative single wavelength lights of 450–650 nm were used for the AQE testing. In the test, time-dependent photocatalytic decomposition of LEV (10 mg/L) by using BTT-DATP-COF was conducted in a photoelectrochemical test system (PEC 2000, Beijing Perfect Light). The degradation text was lasted for 120 min at $T = 25\text{ }^{\circ}\text{C}$ and $\text{pH} = 5.3$ (ultra-pure water condition). A Xenon lamp (Microsolar300, Beijing Perfect light) was equipped as the light source, and bandpass filter (six single wavelength) was vertically placed ~ 10 cm above the reactor. The light irradiation (AM 1.5 G) was about 150 mW/cm^2 , with an illuminated area of $6.08 \times 10^{-3}\text{ m}^2$.

$$AQE = \frac{N_e}{N_p} \times 100\% = \frac{10^9(\nu \times N_A \times K)(h \times c)}{(I \times A \times \lambda)} \times 100\% \quad (5-9)$$

where N_e is the total number of electrons transferred during reaction, N_p is the number of incident photons during reaction, ν is the reaction rate (mol/s), N_A is Avogadro constant ($6.02 \times 10^{23}\text{ mol}^{-1}$). K is the number of participating electrons (number of transferred electrons; it is very hard to accurately qualify the exact transferred electrons during organics degradation due to the variety of formed transformation products (TPs), so in this study, K is selected as 2 for simplified calculation referred to photocatalytic decomposition of H_2O), h is Planck constant ($6.62 \times 10^{-34}\text{ J}\cdot\text{s}$), c is the light speed ($3.0 \times 10^8\text{ m/s}$), I is the light power density of light source (W/m^2), A is the incident light area in the experiment (m^2), λ is the incident light wavelength (nm).

Added in Page 14 of Revised Manuscript, “Besides, the apparent quantum efficiency (AQE) provided by BTT-DATP-COF were evaluated at different single light wavelength of 450–650 nm, and the material displayed a maximum AQE value of 34% at 450 nm (**Fig. S21**), which are well associated with UV-vis DRS spectra.”

Supplementary Figure 21. The apparent quantum efficiency (AQE) of BTT-DATP-COF.

9. The authors should provide error bars for all curves in Figure 2e.

Response: Thanks for the reviewer's issue. We appreciate their attention to detail. We have included error bars in all three types of COFs in **Fig. 2e**, and the error bar for BTT-DATP-COF is very short and therefore not very noticeable.

Fig. 2e Photocatalytic activity for LEV degradation by different COF under visible-light illumination

10. As framework materials, what are the local structural models chosen by the authors for the calculation?

Response: Thanks for the reviewer's issue. The calculation of the whole manuscript is based on periodic structure of cell unit rather than local structure. The relevant periodic structure model, shown in **Fig S38**, is as follows.

Added in Page 26 of Revised Manuscript, "In order to deeply explain the dynamic difference of LEV degradation by the three COFs, DFT calculations on interaction of LEV molecule and COFs are performed, mainly involving the adsorption energy and the effect of spin structures related to electronic states, and the selection of COFs periodic structure is shown in **Fig. S39**."

Supplementary Figure 39. The structural model of three COFs

REVIEWER COMMENTS

Reviewer #1 (Remarks to the Author):

Twistedly Hydrophobic Basis with Suitable Aromatic Metrics in Covalent Organic Networks Govern Micropollutant Decontamination

Chencheng Qin, Yi Yang, Xiaodong Wu, Long Chen, Xingzhong Yuan, Lin Tang, Danlian Huang, Dongbo Wang, Lai Lyu, Wen Liu, and Hou Wang

The authors have addressed well for most of all my concerns in the initial version of the manuscript. The claims on the long term stability for catalytic reactions have been well supported in this version with convincing data-sets. The discussions on the p-p interactions are, I feel, reasonable. No additional experimental evidence has been give in this stage, but I feel the additional statements with proper citations are enough for the readers to trace the logical story. I have tried to check the degree of interaction with simple calculational model, showing the consistent trends to the authors' scenario.

One suggestion is on the evaluation of optical absorption of the COFs. Indeed, optical absorption of COFs have revised in this version of manuscript with remarkable change in spectra, especially for BTT-DAB-COF. It was a bit surprise for me, because not only the signal-to-noise ration, but total shape of the spectra changed dramatically in comparison with the one given initial version of the manuscript. The ratio degrade a bit in this revised version of the manuscript (figure 2a), and now the normalized version of the spectra are overlapped with each other except around the peaks. In the authors' rebuttal letter, the spectra are simply normalized for comparison, but for me, those are the different ones. And thus I am sorry but I could not support the data give herein, including the ones derived from Tauc analysis. I would like to ask the authors to check carefully the data, and recommend to re-examin the spectroscopy results thoroughly.

Reviewer #2 (Remarks to the Author):

My major concerns were addressed to some degree, the following problems still need to be addressed.

- 1, The explanation of the pore distribution at ~ 1.8 nm for the COFs should be added in the revised manuscript or supporting information.
- 2, In comparison with BTT-DAB-COF, I have always believed that there is an obvious blueshift of the UV-visible spectrum for BTTDADP-COF and BTT-DATP-COF. The authors are suggested to observe the changes that take place in their characteristic peaks (~ 500 nm).
- 3, The degradation ratio of BTT-DATP-COF for phenol intermediate is not high (about 65%) at 120 min, while LEV was completely degraded in 80 min. Why?
- 4, Please explain why the re-tested datum of fs-TA is not consistent with the previous one because their difference is very obvious.
- 5, How to measure the microcrystalline size of COF materials? I cannot see the size change of COFs described in the manuscript from the SEM and TEM images (16.80, 25.88 and 32.30 nm).

Reviewer #3 (Remarks to the Author):

The authors have made good efforts on revising the manuscript, which has been improved on the clarity with some scientific flaws addressed. However, concerns still remain on the proposed structural models of the COFs, specifically on the torsion angles and the stacking behaviors. The detailed comments and questions are listed as following.

1. The authors mentioned the torsion angles are chosen because of the previous report (J. Am. Chem. Soc. 2020, 142, 21, 9752–9762). In this previous report, the torsion angles are obtained from single crystal data. In this work, BTT is used instead of TH, resulting a different framework. This will affect the torsion angles. Because torsion angles are a highlight in the manuscript, the authors need to provide a robust measurement from which the torsion angles can be obtained, e.g., single crystal diffraction, or the density map from Rietveld refinement against PXRD data. While computational study can also give insights into the torsion angles, the authors need to further elaborate the accuracy of the method. For example, how much difference in the torsion angle can the method be used to

distinguish them? Does the difference significant enough to result in a change in the COF's performance? In addition, how the torsion angles are determined should be described in the main text.

2. The authors have obtained TEM images to justify the AA stacking of the COFs. However, the scale of the pores in the TEM images is several times smaller than that in the proposed models.

Response to Reviewers: NCOMMS-23-06406A

Reviewer #1 (Remarks to the Author):

Twistedly Hydrophobic Basis with Suitable Aromatic Metrics in Covalent Organic Networks Govern Micropollutant Decontamination

Chencheng Qin, Yi Yang, Xiaodong Wu, Long Chen, Xingzhong Yuan, Lin Tang, Danlian Huang, Dongbo Wang, Lai Lyu, Wen Liu, and Hou Wang

The authors have addressed well for most of all my concerns in the initial version of the manuscript. The claims on the long-term stability for catalytic reactions have been well supported in this version with convincing data-sets. The discussions on the p-p interactions are, I feel, reasonable. No additional experimental evidence has been given in this stage, but I feel the additional statements with proper citations are enough for the readers to trace the logical story. I have tried to check the degree of interaction with simple calculational model, showing the consistent trends to the authors' scenario.

The reviewer's valuable time and effort on our manuscript are greatly appreciated.

One suggestion is on the evaluation of optical absorption of the COFs. Indeed, optical absorption of COFs have revised in this version of manuscript with remarkable change in spectra, especially for BTT-DAB-COF. It was a bit surprise for me, because not only the signal-to-noise ration, but total shape of the spectra changed dramatically in comparison with the one given initial version of the manuscript. The ratio degrade a bit in this revised version of the manuscript (figure 2a), and now the normalized version of the spectra are overlapped with each other except around the peaks. In the authors' rebuttal letter, the spectra are simply normalized for comparison, but for me, those are the different ones. And thus I am sorry but I could not support the data give herein, including the ones derived from Tauc analysis. I would like to ask the authors to check carefully the data, and recommend to re-examin the spectroscopy results thoroughly.

Response: Thanks for the reviewer's issue. It's important to clarify that the **Fig. 2a**, as mentioned by the reviewer, presents the aperture distribution of the three COFs, rather than the normalized figure of UV-vis diffuse reflection spectra. We checked carefully the data and re-examined the spectroscopy results, these data were consistent with the figures provided. Regarding the results before and after normalization (**Fig. R1**), the overall shape of the entire spectrum for the three COFs didn't change significantly, excepting for the consistent highest points.

In semiconductors, the absorption edge determines the band gap value, which further determines the light absorption capacity (*Phys. Chem. B*, 2005, 109, 23, 11442–11449; *J. Phys. Chem. B*, 2006, 110, 42, 20808–20814). Through extensive literature research (*J. Am. Chem. Soc.* 2020, 142, 9752–9762; *Adv. Mater.* 2022, 34, 2110266; *Angew. Chem. Int. Ed.* 2022, 61, e202202328; *Angew. Chem. Int. Ed.* 2022, e202200413; *Energy Environ. Sci.* 2022, 15, 830), it could be known that the evaluation of redshift or blueshift in photocatalytic materials using UV-vis diffuse reflection spectra primarily involves measuring the absorption edge value, rather than a simple comparison of the highest

absorption point. Keeping this in mind, we measured the absorption variations of the three COFs, and these results were consistent with the order obtained from their bandgap values (Fig. R2).

Fig. R1 (a) UV-vis DRS pattern of three COFs and (b) Normalized of UV-vis DRS pattern of three COFs.

Fig. R2 (a) UV-vis DRS pattern of three COFs (with absorption edge) and (b) the Kubelka-Munk-transformed reflectance spectra of three COFs.

Reviewer #2 (Remarks to the Author):

My major concerns were addressed to some degree, the following problems still need to be addressed.

The reviewer's valuable time and effort on our manuscript are greatly appreciated.

1, The explanation of the pore distribution at ~1.8 nm for the COFs should be added in the revised manuscript or supporting information.

Response: In line with the suggestions of the reviewer, the explanation of the pore distribution at ~1.8 nm for the COFs was re-added in the revised Supporting Information.

Added in Page 19-20 of Revised Supporting Information: "In this study, Fig. S17 shows that three COFs exhibited the typical IV isotherm with an H3 hysteresis loop at $P/P_0 = 0.10-0.25$, $0.2-0.25$ and $0.25-0.35$ for BTT-DAB-COF, BTT-DADP-COF and BTT-DATP-COF, respectively, indicating its mesoporous structure. However, there is another steep slope at $P/P_0=0.1$, indicating some micropores structure in COF or micropores structure with pore sizes close to mesoporous. The pore distribution was mainly fitted by non-local density functional theory (NLDFT) based on N_2 adsorption-desorption data, as shown in Fig. 2a. The pore distribution at ~1.8 nm for all the three COFs may be due to that the three COFs are microcrystalline materials. There is existing some crystal defects as well as interstitial void among microcrystalline grains. In addition, there may be some amorphous polymers around the ordered hexagonal COFs. When N_2 adsorption-desorption experiments are conducted, this amorphous polymerization is compacted together. Therefore, there are two different kinds of pores appeared."

2, In comparison with BTT-DAB-COF, I have always believed that there is an obvious blueshift of the UV-visible spectrum for BTTDADP-COF and BTT-DATP-COF. The authors are suggested to observe the changes that take place in their characteristic peaks (~500 nm).

Response: The nice question from the reviewer is really appreciated. In semiconductors, the absorption edge determines the band gap value, which further determines the light absorption capacity (*Phys. Chem. B*, 2005, 109, 23, 11442–11449; *J. Phys. Chem. B*, 2006, 110, 42, 20808–20814). Through extensive literature research (*J. Am. Chem. Soc.* 2020, 142, 9752–9762; *Adv. Mater.* 2022, 34, 2110266; *Angew. Chem. Int. Ed.* 2022, 61, e202202328; *Angew. Chem. Int. Ed.* 2022, e202200413; *Energy Environ. Sci.* 2022, 15, 830), it could be known that the evaluation of redshift or blueshift in photocatalytic materials using UV-vis diffuse reflection spectra primarily involves measuring the absorption edge value, rather than a simple comparison of the highest absorption point. Keeping this in mind, we measured the absorption variations of the three COFs, and these results were consistent with the order obtained from their bandgap values (**Fig. R2**).

Fig. R2 (a) UV-vis DRS pattern of three COFs (with absorption edge) and (b) the Kubelka–Munk-transformed reflectance spectra of three COFs.

3, The degradation ratio of BTT-DATP-COF for phenol intermediate is not high (about 65%) at 120 min, while LEV was completely degraded in 80 min. Why?

Response: Thanks reviewer for raising the nice issue.

On the one hand, the concentration of phenol was set at 10 mg/L in our previous experiment. It's important to note that in the course of LEV degradation, phenol serves as just one of the by-products or intermediates, and its concentration is not typically so high. Besides, the mineralization rate of LEV was only 54% in this work. Therefore, the degradation rate of phenol may be related to the concentration. We have now adjusted the phenol concentration to 1 mg/L, which is a more representative value for its presence during the LEV degradation process. The updated degradation diagram is illustrated in the **Fig. R3**. It can be seen that the degradation rate of phenol by BTT-DATP-COF increases from 54% to 95% when the concentration was reduced to 1 mg/L.

Fig. R3 Photocatalytic activity for 10 mg/L (a) and 1 mg/L (b) phenol degradation by BTT-DATP-COF under visible-light illumination.

On the other hand, for levofloxacin (LEV) degradation, phenol was a deep oxidation product, which had a more stable structure compared with LEV. The reactive oxygen species (ROS)-induced oxidation, the degradation activity of the organic compound is highly related to its chemical structure (*Environ. Sci. Technol.* 2022, 56, 8784-8795). Previous studies have reported that the degradation activity is related to the molecular gap (parameter of “ELUMO–HOMO”) based on quantitative structure-activity relationship

(QSAR) analysis, and a lower molecular gap leads to high reactivity (*Environ. Sci. Technol.* 2015, 49, 13394-13402). In this study, the value of ELUMO–HOMO is obtained to be 4.3 eV for LEV and 6.5 eV for phenol (**Fig. R4**) based on DFT calculation, so LEV can be more easily degraded under the attack of ROS. In addition, LEV holds some highly reactive sites, especially the two N atoms in the piperazine ring (Fig. 3d), leading to efficient degradation of LEV through dealkylation and oxidation for the cleavage of the piperazine ring (*PNAS.* 2023, e2305378120). Therefore, for the degradation of LEV and phenol with the same initial concentration (10 mg/L in this work), the higher degradation rate constant and removal efficiency were exhibited for the LEV.

Fig. R4 The HOMO and LUMO energy level of LEV and phenol calculated on Gaussian 16 using the B3LYP/6-31G* method.

4, Please explain why the re-tested datum of fs-TA is not consistent with the previous one because their difference is very obvious.

Response: We would like to address this issue in detail. Several factors contribute to these observations:

(i) Repeatability and Stability. When utilizing the same COFs material and experimental conditions, minor variations in experimental procedures and environmental factors can lead to discrepancies in results. To address this, we conducted three repetitions of the second measurement (BTT-DAB-COF and BTT-DADP-COF), eventually arriving at the data presented in this paper.

(ii) Diversity in Material Structure and Crystal Forms. The covalent organic framework (COF) materials employed in this study were synthesized via a solvothermal reaction. An important factor determining crystallinity in COFs is the dynamic nature of the bonds used to join the building blocks (*Chem. Soc. Rev.*, 2020, 49, 2291-2302). This feature allows self-healing and error correction of the structures during synthesis. This concept is named “chemically induced reversibility”. The reactions leading to COF formation are limited to those fulfilling the reversible character required to allow the formation of ordered structures. Therefore, these processes can obtain COF materials with distinct crystal morphologies or structural irregularities, leading to varied responses in transient absorption spectra and consequently resulting in different absorption characteristics within the spectra.

(iii) Material Non-Uniformity. Given that our material exists in a powdered form, we

dispersed the COFs material in water for transient absorption spectrum testing to simulate experimental conditions. However, the non-uniform dispersion of COFs material within the solution can also impact test outcomes (*Angew. Chem. Int. Ed.* 2023, e202309480).

5, How to measure the microcrystalline size of COF materials? I cannot see the size change of COFs described in the manuscript from the SEM and TEM images (16.80, 25.88 and 32.30 nm).

Response: Thanks for the reviewer's issue. The microcrystalline size of COF materials in this study have been calculated using XRD test data, the half-peak width, and the Scherrer formula, rather than relying on measurements from SEM and TEM results. The specific calculation process is outlined below:

The microcrystalline size of COF materials in this study have been determined using XRD results, the half-peak width, and the Scherrer formula (*Phys. Rev.* 1939, 56, 978 and *J. Am. Chem. Soc.* 2020, 142, 4862-4871), as opposed to measurements derived from SEM and TEM analyses. The specific calculation procedure is outlined below (Supplementary Note 1):

The full width at half maximum (FWHM) correlates with the crystal size according to the Scherrer equation:

$$D = \frac{K\lambda}{\beta \cos\theta}$$

where D is the mean size of the ordered (crystalline) domains, which may be smaller or equal to the grain size; K is a dimensionless shape factor, with a value close to unity. The shape factor has a typical value of about 0.9, but varies with the actual shape of the crystallite. λ is the X-ray wavelength; β is the line broadening at half the maximum intensity (FWHM), after subtracting the instrumental line broadening, in radians. This quantity is also sometimes denoted as $\Delta(2\theta)$, θ is the Bragg angle.

The (100) peak of BTT-DAB-COF, BTT-DADP-COF and BTT-DATP-COF has a FWHM of 0.47, 0.31 and 0.25, respectively. Take the BTT-DAB-COF for example, $K=0.9$, $\lambda=0.15405$, $\beta=(0.47/180)*3.14$, $2\theta=3.5^\circ$. Therefore, the microcrystal sizes of BTT-DAB-COF are 16.80 nm. By analogy, the microcrystalline sizes of BTT-DADP-COF and BTT-DATP-COF are calculated to 25.88 and 32.30 nm, respectively.

Reviewer #3 (Remarks to the Author):

The authors have made good efforts on revising the manuscript, which has been improved on the clarity with some scientific flaws addressed.

The reviewer's valuable time and effort on our manuscript are greatly appreciated.

However, concerns still remain on the proposed structural models of the COFs, specifically on the torsion angles and the stacking behaviors. The detailed comments and questions are listed as following.

1. The authors mentioned the torsion angles are chosen because of the previous report (J. Am. Chem. Soc. 2020, 142, 21, 9752–9762). In this previous report, the torsion angles are obtained from single crystal data. In this work, BTT is used instead of TH, resulting a different framework. This will affect the torsion angles. Because torsion angles are a highlight in the manuscript, the authors need to provide a robust measurement from which the torsion angles can be obtained, e.g., single crystal diffraction, or the density map from Rietveld refinement against PXRD data. While computational study can also give insights into the torsion angles, the authors need to further elaborate the accuracy of the method. For example, how much difference in the torsion angle can the method be used to distinguish them? Does the difference significant enough to result in a change in the COF's performance? In addition, how the torsion angles are determined should be described in the main text.

Response: Thank you for your valuable feedback on our paper. We fully comprehend the concerns raised by the reviewers regarding our calculations. In response, we have taken a rigorous approach by employing two distinct calculation platforms (Dmol3 package and ORCA quantum chemistry software) with respective calculation methods (DNP basic set and M06-2X functional/def2-TZVPP basis set). Remarkably, both approaches demonstrate the identical trends.

We present the detailed calculation procedures for these two methods: (i) The spin-restricted DFT calculations for the geometry optimized configurations of the three COF units were obtained by Dmol3 package. The generalized gradient approximation (GGA) of the Perdew-Burke-Ernzerhof (PBE) function was applied to describe the exchange-correlation potential, and the double number basis set (DNP) was adopted. The core treatment and TS method for DFT-D correction were adopted by all electron method, and the thresholds for maximum energy, force, and displacement was set to 1.0×10^{-5} Hartree, 2.0×10^{-3} Hartree/Å, and 5.0×10^{-3} Å, respectively; (ii) All-electron DFT calculations have been carried out by the latest version of ORCA quantum chemistry software (Version 5.0.4). The M06-2X functional and def2-TZVPP basis set were adopted for geometry optimization, and the optimal geometry for each molecule was determined. The geometry optimized process by ORCA software can be referred to the animation. The DFT-D3 dispersion correction was applied to correct the weak interaction to improve the calculation accuracy. Furthermore, we have provided the convergence settings employed during the calculation of torsion angles (based on the ORCA package), as depicted in **Fig. R5**. This careful consideration ensures that any computational variations introduced during the process do not exert a substantial impact on the performance evaluation of COFs. Because when we

calculate the torsion angles, we use the unit of the COF material under its free state, and this unit can be restricted by the adjacent molecules when forming the COF material, therefore showing different torsion angles with its free molecules and forming the COF crystals. Finally, we have outlined the complete procedure for calculating torsion angles in our manuscript and the supporting literature.

Convergence Tolerances:			
Energy Change	To1E	1.0000e-06 Eh
Max. Gradient	To1MAXG	1.0000e-04 Eh/bohr
RMS Gradient	To1RMSG	3.0000e-05 Eh/bohr
Max. Displacement	To1MAXD	1.0000e-03 bohr
RMS Displacement	To1RMSD	6.0000e-04 bohr

Fig. R5 The convergence settings employed during the calculation of torsion angles (based on the ORCA quantum chemistry software).

Furthermore, in the previous report (*J. Am. Chem. Soc.* 2020, 142, 21, 9752–9762) the torsion angle of the 4,4-diaminoterphenyl (DATP) monomer at 27° was determined through reference to the single crystal data available. We also obtained single crystal data for 1,4-Diaminobenzene (DAB, CSD number: 1828333) and 4,4-Diaminodiphenyl (DADP, CSD number: 625724), and subsequently assessed their torsion angles within Materials Studio platform. Illustrated in **Fig. R6**, the calculated torsion angles for DAB and DADP are 22.740° and 26.085°, respectively. As a result, the trend in torsion angles as derived from single crystal data concurs with the trend discerned from computational calculations.

Fig. R6 Torsion angle of DAB (left) and DADP (right) obtained by a single crystal.

2. The authors have obtained TEM images to justify the AA stacking of the COFs. However, the scale of the pores in the TEM images is several times smaller than that in the proposed models.

Response: Thanks for the reviewer's issue. We think that owing to the limited conductivity and threshold energy of COFs, TEM testing requires the utilization of a 200 kV high-energy electron beam to attain high-resolution images, which possibly induce the damage. Nonetheless, this elevated-energy electron beam's interaction with the material's surface can induce localized heating, thermal expansion/collapse and even knock-on atomic displacement, subsequently leading to the distortion and deformation of the COF material (*J. Am. Chem. Soc.* 2020, 142, 41, 17224–17235). Consequently, this phenomenon has the potential to introduce disparities in the observed aperture as compared to its proposed models. Moreover, we have conducted a comparison between the experimentally derived

pores from N₂ adsorption-desorption and those presented in the theoretical model. The trends observed in both datasets exhibit alignment. While some distinctions naturally arise due to the calculation-based nature of both approaches, these differences remain well within the acceptable range (*J. Am. Chem. Soc.* 2016, 138, 14, 4710–4713; *J. Am. Chem. Soc.* 2022, 144, 11, 5145–5154; *J. Am. Chem. Soc.* 2022, 144, 6, 2468–2473).

Fig. R6 The pores from N₂ adsorption-desorption isotherm with those presented in the theoretical model for three COFs.

REVIEWERS' COMMENTS

Reviewer #1 (Remarks to the Author):

All my concerns has been addressed and well resolved. Base on the data on the absorption edges, the protocols to determine the ones are in principle, acceptable. One minor issues is not to state on "the order" of the values in the manuscript. All the data suggested the values of bandgaps are "identical", and no meaning to state on the order. This point can be fixed in the proofreading processes, and now I feel the manuscript is ready for publication.

Reviewer #2 (Remarks to the Author):

The authors have addressed my questions, it is recommended for publication in its current form.

Reviewer #3 (Remarks to the Author):

The authors addressed my concern about the torsion angles, but failed to address the concern on the TEM images. A rough estimation of the images in Supplementary Figure 8 resulted in a spacing of 3-4 Å, which is 6-10 times smaller than the pores in the proposed models, i.e., 3.2 nm. The authors attributed the disagreement to electron beam damage. However, electron beam damage usually only transforms COF crystals becoming less crystalline. As far as I know, there is no such way that electron beam can transform COFs into smaller pores and maintaining a good crystallinity. The difference is huge, and it cannot be explained by the errors in the measurement. I suggest the authors to thoroughly check the TEM results. In any case, if the TEM images cannot reflect the true state of the COFs, I suggest the authors to remove the images and revise the related discussions, especially to those related to the claim of AA stacking. Furthermore, the additional discussion on the single crystal analysis of DAB and DADP needs to be implemented in the manuscript.

Response to Reviewers: NCOMMS-23-06406B

Reviewer #1 (Remarks to the Author):

All my concerns has been addressed and well resolved. Base on the data on the absorption edges, the protocols to determine the ones are in principle, acceptable. One minor issues is not to state on "the order" of the values in the manuscript. All the data suggested the values of bandgaps are "identical", and no meaning to state on the order. This point can be fixed in the proofreading processes, and now I feel the manuscript is ready for publication.

Response: The editor's suggestions are really appreciated. We have re-elaborated the relevant content in the revised manuscript.

Added in Page 9 of Revised Manuscript: "By the UV-visible diffuse reflectance spectrum (UV-Vis DRS) in Fig. 2b, three COFs have an absorption onset at 651, 660 and 669 nm, respectively. The corresponding energy band gaps (E_g) decreased from 2.21, to 2.18, and 2.15 eV (Fig. S10)."

Reviewer #2 (Remarks to the Author):

The authors have addressed my questions, it is recommended for publication in its current form.

We thank the reviewer for the time and efforts on our work.

Reviewer #3 (Remarks to the Author):

The authors addressed my concern about the torsion angles, but failed to address the concern on the TEM images. A rough estimation of the images in Supplementary Figure 8 resulted in a spacing of 3-4 Å, which is 6-10 times smaller than the pores in the proposed models, i.e., 3.2 nm. The authors attributed the disagreement to electron beam damage. However, electron beam damage usually only transforms COF crystals becoming less crystalline. As far as I know, there is no such way that electron beam can transform COFs into smaller pores and maintaining a good crystallinity. The difference is huge, and it cannot be explained by the errors in the measurement. I suggest the authors to thoroughly check the TEM results. In any case, if the TEM images cannot reflect the true state of the COFs, I suggest the authors to remove the images and revise the related discussions, especially to those related to the claim of AA stacking. Furthermore, the additional discussion on the single crystal analysis of DAB and DADP needs to be implemented in the manuscript.

Response: The editor's suggestions are really appreciated. In line with the comments, we removed the part of Fig. S8 with the corresponding description of AA stacking. Additional discussion was added on DAB and DADP single crystal analysis in the revised manuscript.

Added in Page 9 of Revised Manuscript: "Besides, the high-resolution TEM (HRTEM) images exhibited the interlayer stacking of 3.9, 3.6 and 3.3 Å along the c direction for respective BTT-DAB-COF (Supplementary Fig. 8a), BTT-DADP-COF (Supplementary

Fig. 8b) and BTT-DATP-COF (Supplementary Fig. 8c), corresponding to the (001) plane. The trend of wide peak value from PXRD pattern is negatively correlated to the interlayer stacking exhibited by the high-resolution TEM (HRTEM) images, indicating a better degree of longitudinal conjugation for BTT-DATP-COF.”

Added in Page 7 of Revised Manuscript: “Furthermore, DAB, DADP and DATP building block have torsion angles via measuring the dihedral angles using adjacent four atoms, corresponding to 28.5°, 145.8° and 146.9° respectively (Fig. 1a). The trend in torsion angles as derived from single crystal data concurs with the trend discerned from computational calculations. The corresponding single crystal data are 22.740°, 26.085° and 27° for DAB (CSD number: 1828333) DADP (and CSD number: 625724) DATP (CSD number: 1269383), respectively.”